# POST-HOC BIAS SCORING IS OPTIMAL FOR FAIR CLASSIFICATION

**Wenlong Chen** [*†]
Imperial College London
`wenlong.chen21@imperial.ac.uk`

**Yegor Klochkov** [*]
ByteDance Research
`yegor.klochkov@bytedance.com`

**Yang Liu**
ByteDance Research
`yang.liu01@bytedance.com`

## ABSTRACT

We consider a binary classification problem under group fairness constraints, which can be one of Demographic Parity (DP), Equalized Opportunity (EOp), or Equalized Odds (EO). We propose an explicit characterization of Bayes optimal classifier under the fairness constraints, which turns out to be a simple modification rule of the unconstrained classifier. Namely, we introduce a novel instance-level measure of bias, which we call *bias score*, and the modification rule is a simple linear rule on top of the finite amount of bias scores. Based on this characterization, we develop a *post-hoc* approach that allows us to adapt to fairness constraints while maintaining high accuracy. In the case of DP and EOp constraints, the modification rule is thresholding a single bias score, while in the case of EO constraints we are required to fit a linear modification rule with 2 parameters. The method can also be applied for composite group-fairness criteria, such as ones involving several sensitive attributes. We achieve competitive or better performance compared to both *in-processing* and *post-processing* methods across three datasets: Adult, COMPAS, and CelebA. Unlike most *post-processing* methods, we do not require access to sensitive attributes during the inference time.

## 1 INTRODUCTION

With ML algorithms being deployed in more and more decision-making applications, it is crucial to ensure fairness in their predictions. Although the debate on what is fairness and how to measure it is ongoing (Caton & Haas, 2023), group fairness measures are utilized in practice due to the simplicity of their verification (Chouldechova, 2017; Hardt et al., 2016a), and they conform to the intuition that predictions should not be biased toward a specific group of the population. In practice, it is desirable to train classifiers satisfying these group fairness constraints while maintaining high accuracy.

Training classifiers that maintain competitive accuracy and satisfy group fairness constraints remains a challenging problem, and it often requires intervention during the training time. A popular approach (Zafar et al., 2017; 2019) suggests relaxing these constraints of a discrete nature to score-based differentiable constraints, thus utilizing gradient-based optimization methods. This approach is very flexible and can be used in a broad set of applications (Donini et al., 2018; Cotter et al., 2019; Rezaei et al., 2021; Wang et al., 2021; Zhu et al., 2023). Another popular method suggests dynamically reweighting observations during training (Agarwal et al., 2018). In vision tasks, researchers propose to use more sophisticated techniques, such as synthetic image generation (Ramaswamy et al., 2021) and contrastive learning (Park et al., 2022).

Another strategy proposes to modify unconstrained classifiers in a *post-hoc* manner (Hardt et al., 2016a; Jiang et al., 2019; Jang et al., 2022). Unlike the *in-processing* methods, these methods allow one to adapt to fairness constraints after the model is trained. These modifications are much cheaper

---

[*]Equal contribution.
[†]Work done during internship at ByteDance Research.

and more feasible in industrial settings, where very large datasets are utilized and complicated algorithms are used to train the target classifier. However, the existing solutions typically require knowledge of the sensitive attribute during the inference time. For instance, Hardt et al. (2016a) propose to modify a score-based classifier in the form $\hat{Y}(X) = \mathbf{1}\{R(X) > t\}$ to a group-specific thresholding rule $\check{Y}(X, A) = \mathbf{1}\{R(X) > t_A\}$. Similar approaches are also taken by Jiang et al. (2019); Jang et al. (2022). This is impractical for real-world applications where sensitive attributes during inference are inaccessible due to privacy protection.

Most of the existing methods aim at debiasing a classifier, whether with an *in-processing* or *post-processing* method. We ask a more general question: how can we flexibly adjust a classifier to achieve the best accuracy for a given level of fairness? For a binary classification problem, Menon & Williamson (2018) study this question from a theoretical perspective, that is, when one knows the ground truth distribution $p(Y, A|X)$, they derive the Bayes-optimal classifier satisfying fairness constraints. Unfortunately, Menon & Williamson (2018) only cover two cases of fairness measures: Demographic Parity and Equalized Opportunity. In this paper, we close this gap and derive the Bayes-optimal classifier for general group fairness metrics, which include the case of Equalized Odds. Our analysis also allows using composite fairness criteria that involve more than one sensitive attribute at the same time. See other related references in Section 5.

We interpret our solution as a modification of (unconstrained) Bayes optimal classifier based on a few values that we term *"bias scores"*, which in turn can be thought of as a measure of bias on instance level. For instance, think of reducing the gender gap in university admissions. Bhattacharya et al. (2017) show that such gap reduction typically happens at the expense of applicants with borderline academic abilities. In terms of classification (passed/ not passed), this corresponds to the group where we are least certain in the evaluation of one's academic abilities. This suggests that evaluation of bias on instance level should not only take into account prediction and group membership, but also uncertainty in the prediction of target value. Our *bias score* not only conforms to this logic, but thanks to being part of Bayes optimal classifier, it is also theoretically principled. In particular, for the case of Demographic Parity constraints, we show that the optimal constrained classifier can be obtained by modifying the output of the unconstrained classifier on instances with largest bias score. When Equalized Odds constraints are imposed, or more generally a composite criterion, the optimal modification is a linear rule with two or more bias scores.

Based on our characterization of the optimal classifier, we develop a practical procedure to adapt any score-based classifier to fairness constraints. In Section 4, we show various experiments across three benchmarks: Adults, COMPAS, and CelebA. We are able to achieve better performance than the in-processing methods, despite only being able to adapt to the group-fairness constraints after training. We also provide competitive results when compared to post-processing methods (Hardt et al., 2016a; Jiang et al., 2019), which require knowledge of sensitive attribute during inference.

We summarize our contributions as follows. Firstly, we characterize the Bayes optimal classifier under group fairness constraints, which generalizes Menon & Williamson (2018) in the sense that Menon & Williamson (2018) can be viewed as a special case in our framework, where the constraint is only a single fairness criterion (e.g. Demographic Parity). Nevertheless, our formulation is more convenient and intuitive thanks to the interpretable *bias score*. Secondly, our characterization can work with composite fairness criterion (e.g. Equalized Odds) as constraint, which has not been established before to our knowledge. Thirdly, based on this characterization, we propose a post-processing method that can flexibly adjust the trade-off between accuracy and fairness and does not require access to test sensitive attributes. Empirically, our method achieves competitive or better performance compared with baselines.

## 1.1 PRELIMINARIES

In this work, we consider binary classification, which consists of many practical applications that motivate machine fairness research (Caton & Haas, 2023). We want to construct a classifier $\hat{Y} = \hat{Y}(X)$ for a target variable $Y \in \{0, 1\}$ based on the input $X$. Apart from the accuracy of a classifier, we are concerned with fairness measurement, given that there is a sensitive attribute $A$, with some underlying population distribution over the triplets $(X, Y, A) \sim \Pr$ in mind. We assume that the sensitive attribute is binary as well. We generally focus on three popular group-fairness criteria:

- **Demographic Parity (DP)** (Chouldechova, 2017) is concerned with equalizing the probability of a positive classifier output in each sensitive group,

$$DP(\hat{Y}; A) = \left| \Pr(\hat{Y} = 1 | A = 0) - \Pr(\hat{Y} = 1 | A = 1) \right|. \tag{1}$$

- **Equalized Odds (EO)** was introduced in Hardt et al. (2016a). Unlike DP which suffers from explicit trade-off between fairness and accuracy in the case where $Y$ and $A$ are correlated, this criterion is concerned with equalizing the false positive and true positive rates in each sensitive group,

$$EO(\hat{Y}; A) = \max_{y=0,1} \left| \Pr(\hat{Y} = 1 | A = 0, Y = y) - \Pr(\hat{Y} = 1 | A = 1, Y = y) \right|. \tag{2}$$

- **Equality of Opportunity (EOp)** was also introduced by Hardt et al. (2016a), and it measures the disparity only between true positive rates in the two sensitive groups,

$$EOp(\hat{Y}; A) = \left| \Pr(\hat{Y} = 1 | A = 0, Y = 1) - \Pr(\hat{Y} = 1 | A = 1, Y = 1) \right|. \tag{3}$$

## 2 BAYES OPTIMAL FAIRNESS-CONSTRAINED CLASSIFIER

For a given distribution over $\Pr \sim (X, Y)$, the Bayes optimal unconstrained classifier has the form $\hat{Y}(X) = \mathbf{1}\{p(Y = 1 | X) > 0.5\}$ in the sense that it achieves maximal accuracy. Although it is not generally possible to know these ground-truth conditional probabilities ($p(Y = 1 | X)$) in practice, such characterization allows one to train probabilistic classifiers, typically with cross-entropy loss. Here we ask what is the optimal classifier when the group fairness restrictions are imposed. Namely, which classifier $\check{Y}(X)$ maximizes the accuracy $Acc(\check{Y}) = \Pr(\check{Y} = Y)$ under the restriction that a particular group fairness measure is below a given level $\delta > 0$.

We are interested in either DP, EOp, or EO constraints, as described in the previous section. For the sake of generality, we consider the following case of *composite criterion*. Suppose, we have several sensitive attributes $A_1, \dots, A_K$ and for each of them we fix values $a_k, b_k$, so that we need to equalize the groups $\{A_k = a_k\}$ and $\{A_k = b_k\}$. In other words, our goal is to minimize a *composite criterion* represented by a maximum over a set of disparities,

$$CC(\check{Y}) = \max_{j=1,\dots,K} \left| \Pr(\check{Y} = 1 | A_k = a_k) - \Pr(\check{Y} = 1 | A_k = b_k) \right|, \tag{4}$$

This general case covers DP, EOp, and EO, as well as composite criteria involving more than one sensitive attribute. Let us give a few examples.

**Example 1** (Demographic Parity). *For the case of DP (see Eq. 1), it is straightforward: take $A_1 = A \in \{0, 1\}$, $a_1 = 0, b_1 = 1$, and then with $K = 1$, $CC(\check{Y}) = DP(\check{Y})$.*

**Example 2** (Equalized Opportunity and Equalized Odds). *Suppose we have a sensitive attribute $A \in \{0, 1\}$, then the Equalized Opportunity criterion (Eq. 2), can be written in the form of Eq. 4 with $A_1 = (A, Y)$, $a_1 = (0, 1)$, and $b_1 = (1, 1)$.*

*For the Equalized Odds, we can write it as a composite criterion with $K = 2$ by setting $A_1 = A_2 = (A, Y)$, and setting $a_1 = (0, 0)$, $b_1 = (1, 0)$, $a_2 = (0, 1)$, $b_2 = (1, 1)$ in Eq. 4.*

**Example 3** (Two and more sensitive attributes). *We could be concerned with fairness with respect to two sensitive attributes $A, B$ simultaneously (for instance, gender and race). In this case, we want to minimize the maximum of two Demographic Parities, which looks as follows*

$$\max\left\{ \left| \Pr(\hat{Y} = 1 | A = 0) - \Pr(\hat{Y} = 1 | A = 1) \right|, \left| \Pr(\hat{Y} = 1 | B = 0) - \Pr(\hat{Y} = 1 | B = 1) \right| \right\}.$$

*If we have three DPs, we will have $K = 3$ in Eq. 4; if we are interested in a maximum over EO's for two different sensitive attributes, we would have $K = 4$, etc.*

Given a specified fairness level $\delta > 0$, we want to find the optimal classifier $\check{Y}(X)$, possibly randomized, that is optimal under the composite criterion constraints

$$\max \quad Acc(\check{Y}) = \Pr(\check{Y} = Y) \qquad \text{s.t.} \qquad CC(\check{Y}) \leq \delta. \tag{5}$$

We will be searching the solution in the form of modification of the Bayes optimal unconstrained classifier. Recall that in our notation, $\hat{Y} = \hat{Y}(X)$ denotes the Bayes optimal unconstrained classifier $\mathbf{1}\{p(1|X) > 0.5\}$. We "reparametrize" the problem by setting $\kappa(X) = \Pr(\check{Y} \neq \hat{Y}|X)$ as the target function. In other words, given an arbitrary function $\kappa(X) \in [0, 1]$, we can define the modification $\check{Y}(X)$ of $\hat{Y}(X)$ by drawing $Z \sim Be(\kappa(X))$ and outputting

$$\check{Y} = \begin{cases} \hat{Y}, & Z = 0 \\ 1 - \hat{Y}, & Z = 1 \end{cases}$$

We call such function $\kappa(X)$ a *modification rule*. With such reparameterization, the accuracy of a modified classifier can be rewritten as

$$Acc(\check{Y}) = Acc(\hat{Y}) - \int \eta(X)\kappa(X)d\Pr(X),$$

where $\eta(X) = 2p(Y = \hat{Y}|X) - 1$ represents the confidence of the Bayes optimal unconstrained classifier $\hat{Y}$ on the instance $X$ (see Section A.1 for detailed derivation), A similar representation holds for the value of the composite criterion. Specifically, recall the criterion is of the form $CC(\check{Y}) = \max_{k \leq K} |C_k(\check{Y})|$, where

$$C_k(\check{Y}) = \Pr(\check{Y} = 1|A_k = a_k) - \Pr(\check{Y} = 1|A_k = b_k).$$

We can rewrite it as

$$C_k(\check{Y}) = C_k(\hat{Y}) - \int f_k(X)\kappa(X)d\Pr(X), \tag{6}$$

$$f_k(X) := (2\hat{Y} - 1)\left[\frac{p(A_k = a_k|X)}{\Pr(A_k = a_k)} - \frac{p(A_k = b_k|X)}{\Pr(A_k = b_k)}\right]. \tag{7}$$

These two expressions suggest that modifying the answer on a point with low confidence $\eta(X)$ makes the least losses in the accuracy, while modifying the answers with higher absolute value of $f_k(X)$ makes largest effect on the parity values, although this has to depend on the sign. This motivates us to define modification rule on the relative score,

$$\text{(Instance-Level Bias Score):} \qquad s_k(X) = \frac{f_k(X)}{\eta(X)}, \tag{8}$$

which we refer to as *bias score*. It turns out that the optimal modification rule, i.e. one corresponding to the constrained optimal classifier in Eq. 5, is a simple linear rule with respect to the given $K$ bias scores. We show this rigorously in the following theorem. We postpone the proof to the appendix, Section A.1.

**Theorem 1.** *Suppose that all functions $f_k, \eta$ are square-integrable and the scores $s_k(X) = f_k(X)/\eta(X)$ have joint continuous distribution. Then, for any $\delta > 0$, there is an optimal solution defined in Eq. 5 that is obtained with a modification rule of the form,*

$$\kappa(X) = \mathbf{1}\left\{\sum_k z_k s_k(X) > 1\right\}. \tag{9}$$

This result suggests that for the case of DP, EOp, or EO, given the ground-truth probabilities $p(Y, A|X)$, we only need to fit 1 parameter for either of Demographic Parity and Equalized Opportunity, which essentially corresponds to finding a threshold, and fit a linear rule in two dimensions for the Equalized Odds. Below we consider each of the three fairness measures in detail.

**Demographic Parity.** In the case of DP constraint, we have a single bias score of the form,

$$s(X) = \frac{1}{\eta(X)}(2\hat{Y} - 1)\left[\frac{p(A = 0|X)}{\Pr(A = 0)} - \frac{p(A = 1|X)}{\Pr(A = 1)}\right], \tag{10}$$

and since there is only one score, the modification rule is a simple threshold rule of this bias score $\kappa(X) = \mathbf{1}\{s(X)/t > 1\}$. We note that $t$ can be positive or negative, depending on which group has the advantage, see Section A.1. This allows one to make linear comparison of fairness on the

instance level. That is, departing from a fairness measure defined on a group level, we derive a bias score that measures fairness on each separate instance. For example, in the context of university admissions (Bhattacharya et al., 2017), our bias score conforms with the following logic: it is more fair to admit a student who has high academic performance (lower $\eta(X)$) than one who has borderline performance (higher $\eta(X)$) even though they are both equally likely to come from the advantageous group (same $f(X)$). We note that the problem of measuring fairness and bias on instance level has recently started to attract attention, see Wang et al. (2022); Yao & Liu (2023).

**Equality of Opportunity.** Here we also have the advantage of having a simple threshold rule, corresponding to the score function,

$$s(X) = \frac{1}{\eta(X)}(2\hat{Y} - 1)\left[\frac{p(A = 0, Y = 1|X)}{\Pr(A = 0, Y = 1)} - \frac{p(A = 1, Y = 1|X)}{\Pr(A = 1, Y = 1)}\right].$$

**Remark 2.1** (Comparison to group-aware thresholding). *Let us consider the case where there is a one-to-one correspondence $A = A(X)$. This is equivalent to the case of observed sensitive attribute, and it is trivial to check that our method turns into a group-aware thresholding and becomes oblivious (Hardt et al., 2016a; Jang et al., 2022) (i.e., the flipping rule doesn't depend on interpretation of individual features $X$ any more). Indeed, in such case we have $p(A = a, Y = 1|X) = p(Y = 1|X)\mathbf{1}\{A(X) = a\}$, therefore $s(X) = \frac{p(Y=1|X)}{2p(Y=1|X)-1}[a_0\mathbf{1}\{A(X) = 0\} + a_1\mathbf{1}\{A(X) = 1\}]$, where $a_0 = 1/\Pr(A = 0, Y = 1) > 0$ and $a_1 = -1/\Pr(A = 1, Y = 1) < 0$. Then for a given $t_\delta$, the final decision rule turns into $\check{Y}(X, A) = \mathbf{1}\{p(Y = 1|X) > t_A\}$, where $t_A = 0.5 + a_A/(4t_\delta - 2a_A)$.*

**Equalized Odds.** Let us consider the case of optimizing under Equalized Odds constraint in detail. In this case, we need to know the ground-truth conditional probabilities $p(Y, A|X)$, and we obtain two scores for $k = 0, 1$,

$$s_k(X) = \frac{1}{\eta(X)}\{2\hat{Y} - 1\}\left[\frac{p(Y = k, A = 0|X)}{\Pr(Y = k, A = 0)} - \frac{p(Y = k, A = 1|X)}{\Pr(Y = k, A = 1)}\right] \tag{11}$$

Our goal is then to find a linear rule in the bias embedding space $(s_0(X), s_1(X))$, which on validation, achieves the targeted equalized odds, while maximizing the accuracy. Notice that here the problem is no longer a simple threshold choice as in the case of DP-constrained classifier. We still need to fit a fairness-constrained classifier, only we have dramatically reduced the complexity of the problem to dimension $K = 2$, and we only have to fit a linear classifier.

We demonstrate the modification rule in the case of EO constraints with the following synthetic data borrowed from Zafar et al. (2019) [Section 5.1.2] as example:

$$p(X|Y = 1, A = 0) = \mathcal{N}([2, 0], [5, 1; 1, 5]), \qquad p(X|Y = 1, A = 1) = \mathcal{N}([2, 3], [5, 1; 1, 5]),$$
$$p(X|Y = 0, A = 0) = \mathcal{N}([-1, -3], [5, 1; 1, 5]), \quad p(X|Y = 0, A = 1) = \mathcal{N}([-1, 0], [5, 1; 1, 5]). \tag{12}$$

We sample 500, 100, 100, 500 points from each of groups $(Y, A) = (1, 0), (1, 1), (0, 0), (0, 1)$, respectively, so that $Y$ and $A$ are correlated. Next, we fit a logistic linear regression with 4 classes to estimate $p(Y, A|X)$ and calculate the scores according to the formulas Eq. 11. In Figure 1a, we show the scatter plot of the scores $(s_0(X), s_1(X))$, with the corresponding group marked by different colors. Figures 1b-1c show the optimal flipping rule, with color encoding $\kappa(X)$ evaluated with the discretized version of the linear program, while the red line approximately shows the optimal linear separation plane. We observe that some of the points that we had to flip for the restriction $EO \le \delta = 0.15$, are unflipped back when the restriction is tightened to $EO \le \delta = 0.01$. It indicates that unlike in the case of DP restriction, there is no unique score measure that can quantify how fair is the decision made by a trained classifier.

## 3 METHODOLOGY

In practice, especially for deep learning models, unconstrained classifiers are usually of the form $\hat{Y} = \mathbf{1}\{\hat{p}(Y|X) > 0.5\}$, with the conditional probability trained using the cross-entropy loss. Our characterization of the optimal modification rule naturally suggests a practical post-processing algorithm that takes fairness restriction into account: assume that we are given an *auxiliary* model for

either $\hat{p}(A|X)$ (in the case of DP constraints) or $\hat{p}(Y, A|X)$ (in the case of EOp and EO constraints). We then treat these estimated conditionals as ground-truth conditional distributions, plugging them into Eq. 8 to compute the bias scores, and modify the prediction $\hat{Y}$ correspondingly with a linear rule over these bias scores. We propose to fit the linear modification rule using a labeled validation set. We call this approach *Modification with Bias Scores* (MBS) and it does not require knowing test set sensitive attribute since the bias scores are computed based on the estimated conditional distributions related to sensitive attribute ($\hat{p}(A|X)$ or $\hat{p}(Y, A|X)$) instead of the empirical observations of sensitive attribute. Here we demonstrate the algorithms in detail for the two cases where DP and EO are the fairness criteria.

**Post-processing algorithm with DP constraints.** In this case, we assume that we have two models $\hat{p}(Y|X)$ and $\hat{p}(A|X)$ (which in the experiments are fitted over the training set, but can be provided by a third party as well) to estimate the ground truth $p(Y|X)$ and $p(A|X)$ respectively. We then define the bias score as follows:

$$\hat{s}(X) = \frac{\hat{f}(X)}{\hat{\eta}(X)} = \frac{\{2\hat{Y}(X) - 1\}\left[\frac{\hat{p}(A=0|X)}{\widehat{\Pr}(A=0)} - \frac{\hat{p}(A=1|X)}{\widehat{\Pr}(A=1)}\right]}{2\hat{p}(Y = \hat{Y}(X)|X) - 1}, \tag{13}$$

where $\widehat{\Pr}(A = i)$ ($i = 0, 1$) can be estimated by computing the ratio of the corresponding group in the training set. We search for the modification rule of the form $\kappa(X) = \mathbf{1}\{\hat{s}(X)/t > 1\}$, so that the resulting $\check{Y}_t(X) = \mathbf{1}\{\hat{s}(X)/t \leq 1\}\hat{Y}(X) + \mathbf{1}\{\hat{s}(X)/t > 1\}(1 - \hat{Y}(X))$ satisfies the DP constraint, while maximizing the accuracy. For this, we assume that we are provided with a labeled validation dataset $\{(X_i, Y_i, A_i)\}_{i=1}^{N_{val}}$ and we choose the threshold $t$ such that the validation accuracy is maximized, while the empirical DP evaluated on it is $\leq \delta$. To find the best threshold value, we simply need to go through all $N_{val}$ candidates $t = \hat{s}(X_i)$, which can be done in $O(N_{val} \log N_{val})$ time. See detailed description in Algorithm 1 in the appendix, Section B.1.

**Post-processing algorithm with EO constraints.** In this case, we require an auxiliary model $\hat{p}(Y, A|X)$ with four classes. This allows us to obtain the 2D estimated bias score $(\hat{s}_0(X), \hat{s}_1(X))$, where

$$\begin{aligned}
\hat{s}_0(X) &= \frac{\hat{f}_0(X)}{\hat{\eta}(X)} = \frac{\{2\hat{Y}(X) - 1\}\left[\frac{\hat{p}(A=0,Y=0|X)}{\widehat{\Pr}(A=0,Y=0)} - \frac{\hat{p}(A=1,Y=0|X)}{\widehat{\Pr}(A=1,Y=0)}\right]}{2\hat{p}(Y = \hat{Y}(X)|X) - 1}, \\
\hat{s}_1(X_i) &= \frac{\hat{f}_1(X)}{\hat{\eta}(X_i)} = \frac{\{2\hat{Y}(X) - 1\}\left[\frac{\hat{p}(A=0,Y=1|X)}{\widehat{\Pr}(A=0,Y=1)} - \frac{\hat{p}(A=1,Y=1|X)}{\widehat{\Pr}(A=1,Y=1)}\right]}{2\hat{p}(Y = \hat{Y}(X)|X) - 1},
\end{aligned} \tag{14}$$

where each of the $\widehat{Pr}(A = a, Y = y)$ is again estimated from training set. We are searching for a linear modification rule $\kappa(X) = \mathbf{1}\{a_0\hat{s}_0(X) + a_1\hat{s}_1(X) > 1\}$ that for a given validation set satisfies the empirical EO constraint while maximizing the validation accuracy. We consider two strategies to choose such a linear rule.

In the first approach, we take a subsample of points $\{(\hat{s}_0(X'_m), \hat{s}_1(X'_m))\}_{m=1}^M$ of size $M \leq N_{val}$ and consider all $M(M-1)/2$ possible linear rules passing through any two of these points. For each of these rules, we evaluate the EO and accuracy on validation set, then choose maximal accuracy among ones satisfying $EO \leq \delta$. The total complexity of this procedure is $O(M^2 N_{val})$. A formal algorithm is summarized in Algorithm 2 in the appendix, Section B.1.

We also consider a simplified version, where we fix a set of $K$ equiangular directions $w = (\cos(2\pi j/K), \sin(2\pi j/K))$ for $j = 0, \ldots, K - 1$. Then, for a score $w_0\hat{s}_0(X) + w_1\hat{s}_1(X)$ we simply need to choose a threshold, following the procedure in the DP case, where we evaluate the EO and accuracy dynamically. The time complexity in $O(K N_{val} \log N_{val})$, see details in Algorithm 3, Section B.1 in the appendix.

**Remark 3.1.** *Note that as functions of $X$, the probabilities $p(Y|X)$ and $p(Y, A|X)$ must agree in order for Theorem 1 to hold, in the sense that $p(Y|X) = p(Y, 0|X) + p(Y, 1|X)$. However, the form of the algorithms itself, which only requires "plug-in" estimators $\hat{p}(Y|X)$ and $\hat{p}(Y, A|X)$ for ground-truth $p(Y|X)$ and $p(Y, A|X)$ respectively, does not require strict agreement between $\hat{p}(Y|X)$ and $\hat{p}(Y, A|X)$ for it to run. In practice, we can employ $\hat{p}(Y|X)$, $\hat{p}(Y, A|X)$ that were trained separately, and we can run the algorithm even in the case where the auxiliary model $\hat{p}(Y, A|X)$ was*

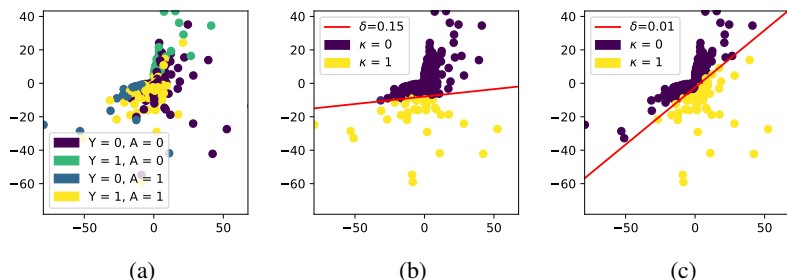

Figure 1: (a) Scatter plot of the scores for synthetic distribution Eq. 12. (b) Separation plane for the optimal flipping rule $\kappa$ corresponding to $EO \leq \delta = 0.15$ and (c) $\delta = 0.01$.

*pretrained by a third party on another dataset similar to the one of interest. E.g., we conduct an experiment where the auxiliary model is based on CLIP (Radford et al., 2021) in Section C.3.*

**Sensitivity analysis.** Here we investigate the sensitivity of our method to inaccuracy in the estimated conditional distributions. We first provide theoretical analysis, which takes into account two sources of error: approximation of the ground-truth conditional distributions $p(Y|X)$ and $p(A|X)$ by, say parametric models, and the sampling error in evaluation of the accuracy and DP on validation. The details regarding the conditions and omitted constants are deferred to the appendix. See precise formulation and the proof in Section E, which also includes general composite criterion.

**Theorem** (Informal). *Suppose that we have estimations of conditional distributions $\hat{p}(Y|X)$, $\hat{p}(A|X)$ (or $\hat{p}(Y, A|X)$ when using EO constraint), and assume that*

$$\mathbb{E}|\hat{p}(Y|X) - p(Y|X)| \leq \varepsilon, \qquad \mathbb{E}|\hat{p}(A|X) - p(A|X)| \leq \varepsilon \qquad \text{(for DP)},$$
$$\mathbb{E}|\hat{p}(Y, A|X) - p(Y, A|X)| \leq \varepsilon \qquad \text{(for EO)}.$$

*Let $\check{Y}$ be the algorithm obtained with Algorithm 1 (Algorithm 2 for EO). Then, with high probability*

$$Acc(\hat{Y}) - Acc(\check{Y}) \lesssim \sqrt{\varepsilon} + \sqrt{(\log N_{val})/N_{val}}, \qquad DP(\check{Y}) - \delta \lesssim \sqrt{(\log N_{val})/N_{val}},$$
$$\left(EO(\check{Y}) - \delta \lesssim \sqrt{(\log N_{val})/N_{val}}\right).$$

Moreover, we include three ablation studies in appendix C, where less accurate $\hat{p}(Y|X)$, $\hat{p}(A|X)$ or $\hat{p}(Y, A|X)$ are deployed to examine the robustness of the post-processing modification algorithm. We find that our post-processing algorithm still retains the performance even when $\hat{p}(Y|X)$, $\hat{p}(A|X)$ or $\hat{p}(Y, A|X)$ are moderately inaccurate.

## 4 EXPERIMENTS

We evaluate MBS on real-world binary classification tasks with the following experimental set-up.

**Datasets.** We consider three benchmarks:

- **Adult Census** (Kohavi, 1996), a UCI tabular dataset where the task is to predict whether the annual income of an individual is above $50,000. We randomly split the dataset into a training, validation and test set with 30000, 5000 and 10222 instances respectively. We pre-process the features according to Lundberg & Lee (2017) and the resulting input $X$ is a 108-dimensional vector. We use "Gender" as the sensitive attribute;

- **COMPAS** (Angwin et al., 2015), a tabular dataset where the task is to predict the recidivism of criminals. The dataset is randomly split into a training, validation and test set with 3166, 1056 and 1056 instances respectively. The input $X$ consists of 9 normal features (e.g. age and criminal history) and we choose "Race" as the sensitive attribute;

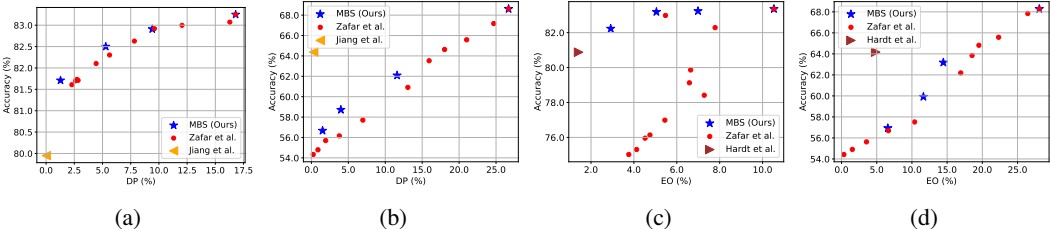

Figure 2: Accuracy (%) vs Demographic Parity (DP) (%) trade-offs on (a) Adult Census and (b) COMPAS; Accuracy (%) vs Equalized Odds (EO) (%) trade-offs on (c) Adult Census and (d) COMPAS. Desired $\delta = \infty$ (unconstrained), $10\%$, $5\%$, and $1\%$.

- **CelebA** (Liu et al., 2015), a facial image dataset containing 200k instances each with 40 binary attribute annotations. We follow the experimental setting as in Park et al. (2022): we choose "Attractive", "Big nose", and "Bag Under Eyes" as target attributes, and choose "Male" and "Young" as sensitive attributes, yielding 6 tasks in total, and we use the original train-validation-test split.

**Network architectures and hyperparameters.** We use an MLP for Adult Census and COMPAS datasets, with hidden dimension chosen to be 8 and 16 respectively. For each CelebA experiment, we use a ResNet-18 (He et al., 2016). For experiments with DP constraints, we train two models $\hat{p}(Y|X)$ and $\hat{p}(A|X)$ to predict the target and sensitive attributes respectively, while for experiments with EO constraints, we only train one model but with four classes $\hat{p}(Y, A|X)$, with each class corresponding to one element in the Cartesian product of target and sensitive attributes.

**Baselines.** For experiments on Adult Census and COMPAS, we compare MBS with Zafar et al. (2017) (Z17), Jiang et al. (2019) (J19) (post-processing version, for experiments with DP constraints) and Hardt et al. (2016a) (H16) (for experiments with EO constraints). For CelebA, we additionally compare with Park et al. (2022) (P22), which is a strong baseline tailored to fair facial attribute classification on CelebA. We report the averaged performance from 3 independent runs for all methods.

**Evaluations & metrics.** We consider both Demographic Parity (DP) and Equalized Odds (EO) as fairness criteria. We select the modification rules $\kappa(X)$ over the validation set according to the algorithms in Section 3. We consider three levels of constraints for the fairness criteria: $\delta = 10\%, 5\%,$ and $1\%$, and we set $M$ in Algorithm 2 described in Section 3 to be 3000, 600 and 5000 for experiments with EO as fairness criterion on Adult Census, COMPAS and CelebA, respectively. Then we report the test set accuracy and DP/EO computed based on the post-processed test predictions after modification according to $\kappa(X)$.

## 4.1 EXPERIMENTS WITH DP AS FAIRNESS CRITERION

We consider experiments with DP as fairness criterion on Adult Census and COMPAS datasets, and we compare MBS with Z17 and J19 (J19). The results are reported in Figures 6a-2b. One can see MBS consistently outperforms Z17 for both datasets in the sense that given different desired levels ($\delta$'s) of DP, MBS tends to achieve higher accuracy. Furthermore, while Z17 requires retraining the model each time to achieve a different trade-off between accuracy and DP, we are able to flexibly balance between accuracy and DP by simply modifying predictions of a single base model according to different thresholds of the bias score. For Adult Census, the $\kappa(X)$ estimated over validation set is robust when evaluated over test set since the DPs for test set are either below the desired $\delta$'s or close to it. For COMPAS, the performance seems to be relatively low as one can see a relatively large drop in accuracy when DP is reduced. Although MBS still outperforms Z17 in this case, it achieves worse performance than J19 and since the validation set is small (1056 instances), the $\kappa(X)$ estimated over it is not robust as there is a relatively big gap between DPs on test set and the specified $\delta$'s. The decline of performance on COMPAS is not surprising since COMPAS is a small dataset (with only 5278 instances), and the number of training examples is insufficient for reliable estimation of both $p(Y|X)$ and $p(A|X)$.

Table 1: Test accuracy (ACC) (%) and EO (%) on CelebA for all six combinations of target and sensitive attributes: (a, m), (a, y), (b, m), (b, y), (e, m), and (e, y) under different desired levels ($\delta$'s) of EO (%) constraints. **Boldface** is used when MBS is better than P22 both in terms of accuracy and EO.

| $\delta$ | (a, m) | | (a, y) | | (b, m) | | (b, y) | | (e, m) | | (e, y) | |
| --- | --- | --- | --- | --- | --- | --- | --- | --- | --- | --- | --- | --- |
| | ACC | EO | ACC | EO | ACC | EO | ACC | EO | ACC | EO | ACC | EO |
| $\infty$ | 81.9 | 23.8 | 82.0 | 24.7 | 84.1 | 44.9 | 84.5 | 21.0 | 85.2 | 19.0 | 85.6 | 13.7 |
| 10 | 80.9 | 6.7 | **79.7** | **10.9** | 83.5 | 8.4 | 84.3 | 10.6 | 85.0 | 8.2 | 85.1 | 9.5 |
| 5 | **80.1** | **2.2** | 78.2 | 6.6 | **83.3** | **4.3** | **84.2** | **4.1** | 84.7 | 5.6 | 85.1 | 5.4 |
| 1 | 79.3 | 4.0 | 76.9 | 2.0 | 83.2 | 1.7 | 83.6 | 2.4 | 83.1 | 1.7 | 83.6 | 1.8 |
| H16 | 71.7 | 3.8 | 71.4 | 3.4 | 78.2 | 0.5 | 80.1 | 1.3 | 80.2 | 1.7 | 81.7 | 1.6 |
| P22 | 79.1 | 6.5 | 79.1 | 12.4 | 82.9 | 5.0 | 84.1 | 4.8 | 83.4 | 3.0 | 83.5 | 1.6 |

## 4.2 EXPERIMENTS WITH EO AS FAIRNESS CRITERION

To evaluate the performance of MBS when the fairness criterion is EO, we again consider Adult Census and COMPAS datasets and the results are reported in Figures 6b-2d. The observation is similar to that in experiments with DP constraints. We again achieve better trade-off between accuracy and EO than Z17 for both datasets. Although H16 can also significantly reduce EO, similar to J19 in the DP-based experiments, it is not able to adjust the balance between EO and accuracy and thus is less flexible than MBS. Both MBS and Z17 achieve relatively low performance for small dataset COMPAS, which we again believe it is due to unreliable estimation of $p(Y, A|X)$ with small dataset. Similar to J19, H16assumes access to the sensitive in test, and thus will not be affected by unreliable inference of the test sensitive attribute.

In addition, we evaluate MBS on a more challenging dataset, CelebA, following the same set-up as in P22. Here we denote the target attributes "Attractive", "Big_Nose", "Bag_Under_Eyes" as "a", "b" and "e" respectively, and the sensitive attributes "Male" and "Young" as "m" and "y" respectively. The results are reported in Table 1. MBS tends to achieve better trade-off than H16, whose accuracy is severely hurt across all 6 tasks. MBS is able to maintain high accuracy and meanwhile achieve competitive or even smaller EO. Furthermore, MBS consistently achieves better or competitive performance when compared with P22. To our knowledge, their method is one of the state-of-the-art methods for fair learning on CelebA. We additionally report validation metrics and standard error across 3 independent runs in Appendix D.

## 5 RELATED LITERATURE

We mention some recent work on Bayes optimal classifiers under approximate fairness constraints. Menon & Williamson (2018) consider cost-aware binary classification, followed by a series of results that characterize sensitive attribute-aware fair classifiers. These include DP constraints with either binary or multi-class target and either binary or multi-class sensitive attribute Denis et al. (2021); Xian et al. (2023); Gaucher et al. (2023). Zeng et al. (2022) additionally considers other group fairness metrics. These papers do not cover the cases of Equalized Odds and Composite Criterion. Implementations in Denis et al. (2021); Zeng et al. (2022) are based on closed form expressions of the threshold weights, while our MBS treats the bias scores as features.

## 6 CONCLUSION

To the best of our knowledge, we have for the first time characterized the Bayes optimal binary classifier under composite group fairness constraints, with a post-hoc modification procedure applied to an unconstrained Bayes optimal classifier. Our result applies to popular fairness metrics, such as DP, EOp, and EO as special cases. Based on this characterization, we propose a simple and effective post-processing method, MBS, which allows us to freely adjust the trade-off between accuracy and fairness. Moreover, MBS does not require test sensitive attribute, which significantly broadens its application in real-world problems where sensitive attribute is not provided during inference.

## ACKNOWLEDGMENTS

We are grateful to the anonymous referees for valuable feedback that helped to improve the presentation, extend experiments and theoretical results. We are thankful to our former collegue Muhammad Faaiz Taufiq for plenty of fruitful conversations. We also thank Yingzhen Li for her valuable comments on the manuscript.

## REPRODUCIBILITY STATEMENT

Details for the experimental set-up are provided in the beginning of Section 4, and the code can be found at https://github.com/chenw20/BiasScore.

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

# A PROOF OF THEOREM 1

## A.1 SIMPLE PROOF FOR THE CASE OF DEMOGRAPHIC PARITY

For the sake of exposition, we first show a simple proof for the case of demographic parity. Observe that,

$$
\begin{aligned}
Acc(\check{Y}) =&(\text{not modifying term}) \int (1 - \kappa(X))p(Y = \hat{Y}|X)d\Pr(X) \\
&+ (\text{modifying term}) \int \kappa(X)(1 - p(Y = \hat{Y}(X)|X))d\Pr(X) \\
=& Acc(\hat{Y}) + \int \kappa(X)[1 - 2p(Y = \hat{Y}(X)|X)]d\Pr(X) \\
=& Acc(\hat{Y}) - \int \kappa(X)\eta(X)d\Pr(X),
\end{aligned}
$$

Let us now write how the demographic parity changes after we modify the answers. We use notation $DP_\pm$ to denote a signed demographic parity that measures difference between positive output for groups $A = 0$ and $A = 1$, so that the actual $DP$ is the absolute value of $DP_\pm$. We have,

$$
\begin{aligned}
DP_\pm(\check{Y}) =& \int \Pr(\check{Y} = 1|X) \left[ \frac{p(A = 0|X)}{\Pr(A = 0)} - \frac{p(A = 1|X)}{\Pr(A = 1)} \right] d\Pr(X) \\
=& \int \left( \hat{Y} + \kappa(X)\{1 - 2\hat{Y}\} \right) \left[ \frac{p(A = 0|X)}{\Pr(A = 0)} - \frac{p(A = 1|X)}{\Pr(A = 1)} \right] d\Pr(X) \\
=& DP_\pm(\hat{Y}) - \int \kappa(X)\{2\hat{Y} - 1\} \left[ \frac{p(A = 0|X)}{\Pr(A = 0)} - \frac{p(A = 1|X)}{\Pr(A = 1)} \right] d\Pr(X),
\end{aligned}
$$

here $DP_\pm(\hat{Y})$ corresponds to the parity of the original classifier $\hat{Y}$. W.l.o.g., assuming that $DP_\pm(\hat{Y})$ is positive (i.e., $A = 0$ is the advantaged group based on the unfair $\hat{Y}$), to decrease the DP maximally, we should sooner reject points with higher score

$$
f(X) = \{2\hat{Y} - 1\} \left[ \frac{p(A = 0|X)}{\Pr(A = 0)} - \frac{p(A = 1|X)}{\Pr(A = 1)} \right],
$$

It suggests when $\hat{Y}(X) = 1$, we should sooner modify the prediction (from 1 to 0) for $X$ that is more likely to come from the advantaged group favoured by the original unfair classifier, and when $\hat{Y}(X) = 1$, we should sooner modify the prediction (from 0 to 1) for $X$ that is more likely to come from the disadvantaged group based on the original unfair classifier. Similarly, if $DP_\pm^*$ is negative, we should modify those points with lowest score. Notice that we always should modify points with sign equal to that of $DP_\pm^*$, unless it is not possible, then we have reached ideal parity $DP_\pm(\kappa) = 0$. Recall that in order to reduce the accuracy the least, we should modify on the points with lowest confidence $\eta(X)$. The relative change then is quantified by the ratio $s(X) = f(X)/\eta(X)$. Turns out, we can prove that the optimal rule correponds to $\kappa(X) = \mathbf{1}\{s(X) > t\}$ (when $DP_\pm(\hat{Y}) > 0$, otherwise we should look for $\kappa(X) = \mathbf{1}\{s(X) < t\}$) by repeating the steps of famous Neyman-Pearson lemma, where we replace the probability of null and alternative hypothesis with $\eta(X)$, $f(X)$, respectively.

Indeed, by contradiction, assume there is a separate rule $\kappa'$ that achieves the same accuracy but smaller $DP_\pm$, so we have that

$$
\int (\kappa'(X) - \kappa(X))\eta(X)d\Pr(X) = 0, \text{ and } \int (\kappa'(X) - \kappa(X))f(X)d\Pr(X) < 0
$$

We assume that $f(X)$ is non-negative everywhere where both $\kappa(X)$ and $\kappa'(X)$ are non-negative, otherwise it corresponds to modifying a good answer on points from disadvantaged group. Let us denote $m(X) = (\kappa'(X) - \kappa(X))\eta(X)$ and $r(X) = f(X)/\eta(X)$. Then, we have that where $m(X) < 0$ we have $r(X) \geq s$ and where $m(X) > 0$ we have $r(X) < s$. So, we weight negative

points with a smaller and positive score, hence contradiction. More, rigorously,

$$
\int m(X)r(X)d\Pr(X) = \int m(X)r(X)\mathbb{1}(m(X) > 0)d\Pr(X)
$$
$$
+ \int m(X)r(X)\mathbb{1}(m(X) < 0)d\Pr(X)
$$
$$
\geq \int m(X)s\mathbb{1}(m(X) > 0)d\Pr(X) + \int m(X)s\mathbb{1}(m(X) < 0)d\Pr(X)
$$
$$
= 0 \,.
$$

### A.2 COMPLETE PROOF OF THEOREM 1

First, let us expand the derivation in Eq. 6. We have,

$$
C_k(\check{Y}) = \int \Pr(\check{Y} = 1|X)\left[\frac{p(A_k = a_k|X)}{\Pr(A_k = a_k)} - \frac{p(A_k = b_k|X)}{\Pr(A_k = b_k)}\right]d\Pr(X)
$$
$$
= C_k^* - \int \kappa(X)\{2\mathbb{1}[\hat{Y} = 1] - 1\}\left[\frac{p(A_k = a_k|X)}{\Pr(A_k = a_k)} - \frac{p(A_k = b_k|X)}{\Pr(A_k = b_k)}\right]d\Pr(X)
$$
$$
= C_k^* - \int \kappa(X)f_k(X)d\Pr(X),
$$

where we set $C_k^* = C_k(\hat{Y})$. Therefore, our optimization problem can be formulated as a linear problem on functions $\kappa(X)$

$$
Acc(\hat{Y}) - Acc(\check{Y}) = \int \eta(X)\kappa(X)d\Pr(X) \to \min_{\kappa(X)\in[0,1]},
$$
$$
\text{s. t.}
$$
$$
\int f_k(X)\kappa(X)d\Pr(X) \leq C_k^* + \delta
$$
$$
- \int f_k(X)\kappa(X)d\Pr(X) \leq -C_k^* + \delta
$$

To prove Theorem 1, it is left to apply the following technical lemma.

**Lemma 1.** *Suppose, we have functions $f_1(X), \ldots, f_K(X), \eta(X)$, and a probability measure $P$ over $X \in \mathcal{X}$. Consider the optimization problem over measurable functions $\kappa(X)$ taking values in $[0, 1]$,*
$$
\langle\kappa, \eta\rangle \to \min, \quad s.t. \quad \langle\kappa, f_k\rangle \leq b_k, \quad k = 1, \ldots, K,
$$
*where $\langle f, g\rangle = \int f(X)g(X)dP(X)$. Suppose, the following conditions hold:*

    A. *all functions $f_k, \eta$ are from $L_2(P)$;*

    B. *there is a strictly feasible solution, in the sense that there is a function $\kappa'(X) \in [0, 1]$ such that $\langle\kappa', f_k\rangle < b_k$;*

    C. *for any $z \in \mathbb{R}_+^k$, we have that $P(\sum_k z_k f_k(X) + \eta(X) = 0) = 0$. Notice that this is a weaker version of the condition that $(f_1/\eta, \ldots, f_K/\eta)$ has continuous distribution.*

*Then, there is $z \in \mathbb{R}_+^K$ and an optimal solution of the*

$$
\kappa(X) = \mathbb{1}\left[\sum_k z_k f_k(X) > \eta(X)\right].
$$

*Furthermore, $z \in \mathbb{R}_+^k$ can be any solution to the problem $\min_{z\geq 0} b^\top z + \int(z^\top F(x) + \eta(x))_+ dP(x)$.*

Notice, that thanks to $\delta > 0$, the restrictions can be met with strict inequalities by the modification rule $\kappa(X) = \mathbb{1}[p(Y = 1|X) < 0.5]$ which corresponds to the constant classifier $\check{Y} = 1$. Furthermore, we note that we have in total $2K$ restrictions, one for each of functions $f_k, -f_k$), and the resulting vector $z$ in Eq. 9 therefore can have negative values, unlike the vector $z$ in the above lemma.

**Remark A.1.** *Zeng et al. (2022) refer to the* generalized Neyman-Pearson lemma*, which looks very similar to the lemma above. In their formulation however, it is required that such $z_1, \ldots, z_k$ exist that turn all inequality constraints into equalities. We avoid this requirement by using linear programming duality. In general, the optimal $z_1, \ldots, z_k$ do not have to satisfy all constraints as equality, which extends the range of cases where our lemma can be applied.*

*Proof of Lemma 1.* Consider an operator $F : L_2(P) \mapsto \mathbb{R}^K$ such that $F\kappa = (\langle f_1, \kappa \rangle, \ldots, \langle f_K, \kappa \rangle)^\top$. Below we write that a vector $x$ satisfies $x \geq 0$ if all its coordinates are non-negative, i.e. $x \in \mathbb{R}^d_+$ for some $d$. With some abuse of notation we will also say that $z^\top F = \sum_k z_k f_k$. Denote $S_+$ the set of $L_2$ functions with pointwise non-negative values. Our problem reads as follows,

$$\max_\kappa -\langle \eta, \kappa \rangle \qquad \text{s.t.} \qquad \kappa \in S_+, \ 1 - \kappa \in S_+, \ F\kappa \leq b.$$

Consider the dual problem,

$$\min_{z, \lambda} z^\top b + \langle \lambda, 1 \rangle, \qquad \text{s.t.} \qquad z \in \mathbb{R}^K_+, \lambda \in S_+, z^\top F + \lambda + \eta \in S_+. \tag{15}$$

We have the duality property, for any feasible $\kappa$ to primal and $z, \lambda$ to dual, we have

$$-\langle \eta, \kappa \rangle \leq z^\top F \kappa + \langle \lambda, \kappa \rangle \leq z^\top b + \langle \lambda, 1 \rangle.$$

In the linear programming these inequalities are called *slackness condition*. Assume for a moment, that strong duality holds, i.e. there is $\kappa$ and $z, \lambda$ such that all inequalities turn into equalities. Then, both pairs are $\kappa$ and $z, \lambda$ are optimal for their linear programs. Furthermore,

$$\langle \lambda, 1 - \kappa \rangle = 0, \qquad \langle \kappa, z^\top F + \eta + \lambda \rangle = 0$$

Furthermore, it is straightforward to see that the optimal solution should satisfy almost everywhere $\lambda = (F^\top z + \eta)_-$, where for a function $g(X)$ we denote its negative part $g_-(X) = \max(0, -g(X))$. Simply observe that we must minimize each $\lambda(X)$ independently, while satisfying the condition $z^\top F(X) + \eta(X) + \lambda(X) \geq 0$. Therefore, we have that the set $\{X : z^\top F + \eta + \lambda > 0\}$ is the same as $\{X : z^\top F + \eta > 0\}$, up to a difference of probability 0. So, we can say that $z^\top F + \eta > 0$ yields $\kappa = 1$ $P$-a.s. and $z^\top F + \eta < 0$ yields $\kappa = 0$ $P$-a.s. Assuming $P(z^\top F + \eta = 0) = 0$, we have that there is an optimal solution that has a form $\kappa = \mathbf{1}\{z^\top F + \eta > 0\}$.

Now let us show the strong duality property. Consider the value $v^* = \min_{z \geq 0} z^\top b + \langle (z^\top F + \eta)_-, 1 \rangle$, which is the optimal value in the dual problem, thanks to the identity for the optimal $\lambda$ given $z$. Thanks to its closed form, we show that it is stable w.r.t. the perturbations in the inputs $F, \eta$, so that we can reduce our problem to finite dimensional linear programming (LP), where strong duality is known to hold.

**Step 1 (bounded $z$).** First let us show that any optimal $z$ is bounded. Since for the primal problem the feasible set has non-empty interior, we have that for every $a \in \mathbb{R}^K_+$

$$a^\top b + \langle (a^\top F)_-, 1 \rangle > 0,$$

and notice that this is a continuous function of $a \in \mathbb{R}^K_+$ (follows from $f_k \in L_1$). Therefore, the following number is strictly positive

$$\mathcal{Z}_F := \inf_{\|a\|=1, a \geq 0} a^\top b + \langle (a^\top F)_-, 1 \rangle > 0. \tag{16}$$

Let us show by contradiction that any optimal solution $z \geq 0$ to dual problem satisfies $\|z\| \leq \langle |\eta|, 1 \rangle / \mathcal{Z}_F$. Assume $\|z\| > \langle |\eta|, 1 \rangle / \mathcal{Z}_F$. We have that for $z = La$ with $\|a\| = 1$,

$$La^\top b + \langle (La^\top F + \eta)_-, 1 \rangle \geq L(a^\top b + \langle (a^\top F), 1 \rangle) - \langle \eta_+, 1 \rangle > \langle \eta_-, 1 \rangle$$

as long as $L > \langle |\eta|, 1 \rangle / \mathcal{Z}_F$. Since the objective value $\langle \eta_-, 1 \rangle$ can be achieved with $z = 0$ in the dual problem, so an optimal $z$ must have a norm smaller or equal to $\langle |\eta|, 1 \rangle / \mathcal{Z}_F$.

**Step 2 (discrete approximation).** Next, consider step approximations of $F$ and $\eta$ in the following form. Select some partition $I_1 \cup \cdots \cup I_M = \mathcal{X}$, and set $\tilde{f}_k(X) = \sum_m \tilde{f}_{km} \mathbf{1}\{X \in I_m\}$, $\tilde{\eta}(X) = \sum_m \tilde{\eta}_m \mathbf{1}\{X \in I_m\}$, in a way that $\|\eta - \tilde{\eta}\|_{L_2} \leq \varepsilon$ and $\|f_k - \tilde{f}_k\|_{L_2} \leq \varepsilon$. The fact that $f_k, \eta \in L_1$ is

enough to find such approximation, however, since we require $f_k, \eta \in L_2$ we can also assume that $\tilde{\eta}_k = \mathbb{E}[\eta(X)|X \in I_m]$, $\tilde{f}_{km} = \mathbb{E}[f_k(X)|X \in I_m]$, where each $I_m$ is assumed to have non-zero probability. Let us consider,

$$\tilde{v} = \min_{z \geq 0} z^\top b + \langle (z^\top \tilde{F} + \tilde{\eta})_-, 1 \rangle$$

An optimal $\tilde{z}$ to this problem is bounded by $(\langle |\eta|, 1 \rangle + \varepsilon)/(\mathcal{Z}_F - k\varepsilon) \leq 2\langle |\eta|, 1 \rangle / \mathcal{Z}_F$ for sufficiently small $\varepsilon$. We therefore can assume that both problems are minimized in $[0, D]^K$ instead of $\mathbb{R}_+^K$ for some $D$ that does not depend on (small enough) $\varepsilon$.

We have then (denoting by $\tilde{y}$ an optimal solution for $\tilde{v}$) that

$$\tilde{v} = \tilde{z}^\top b + \langle (\tilde{z}^\top \tilde{F} + \tilde{\eta})_-, 1 \rangle \geq \tilde{z}^\top b + \langle (\tilde{z}^\top F + \tilde{\eta})_-, 1 \rangle - \langle |\tilde{z}^\top (F - \tilde{F})|, 1 \rangle \geq v^* - DK\varepsilon.$$

Similarly, $v^* \geq \tilde{v} - DK\varepsilon$. Let us look at the approximation of the primal problem. Let $u^*$ and $\tilde{u}$ correspondingly be the optimal values of the objective in the original and discrete problem. Let us take the discrete function $\tilde{\kappa}(X) = \sum_m \tilde{\kappa}_m \mathbf{1}\{X \in I_m\}$, where $\tilde{\kappa}_m$ is the elements of the vector that solves the discrete problem. Then, $u^* \geq -\langle \tilde{\kappa}, \eta \rangle = -\langle \tilde{\kappa}, \eta \rangle = \tilde{u}$, where we used the fact that $\tilde{\eta}$ averages $\eta$ over each interval $I_m$. Thanks to the strong duality of LP, for the discrete problem we have that $\tilde{u} = \tilde{v}$. Thus, we have shown that $u^* \geq \tilde{u} = \tilde{v} \geq v^* - DK\varepsilon$, while $u^* \leq v^*$ is already guaranteed by weak duality. Sending $\varepsilon$ to zero proves the required strong duality property. $\quad\square$

# B   PRACTICAL ALGORITHMS

## B.1   ALGORITHMS

Here we present the detailed algorithms we used in our experiments for DP and EO, particularly, for the choice of linear rule. In each case, we assume that a validation set $\{(X_i, Y_i, A_i)\}_{i=1}^{N_{val}}$ is given, based on which we choose the rule satisfying empirical fairness restrictions, whilst maximizing the empirical accuracy.

For the case of DP restrictions, we assume that we are given models $\hat{p}(Y|X), \hat{p}(A|X)$, as well as evaluated probabilities $\widehat{\Pr}(A = 0), \widehat{\Pr}(A = 1)$, which we can either evaluate on a bigger training dataset (in case we have such access), or on the validation dataset. After we calculate the corresponding scores $\hat{s}(X_i)$, we consider rearrangement $\hat{s}(X_{i_1}) \geq \cdots \geq \hat{s}(X_{i_{N_{val}}})$ and test the thresholds $t = \hat{s}(X_{i_j})$ one after the other. When we move to the next candidate $t = \hat{s}(X_{i_{j+1}})$ we only need to spend $O(1)$ time updating the current empirical DP and accuracy. See detailed procedure in Algorithm 1.

For the case of EO restrictions, we assume that we are given models $\hat{p}(Y|X), \hat{p}(Y, A|X)$, which are two functions that do not necessarily agree as probability distributions. We also assume that we are given evaluated probabilities $\widehat{\Pr}(Y = y, A = a), a, y \in \{0, 1\}$. Then, we calculate the corresponding scores $\hat{s}_{0,1}(X_i)$. In order to choose the linear rule of the form $\kappa(X) = \mathbf{1}\{w_0 \hat{s}_0(X) + w_1 \hat{s}_1(X) > t\}$ we consider two strategies. In the first one, we observe that we can restrict the search only to lines that pass through two validation points, which exhaust the search of all possible pairs of accuracy and EO on validation. This, however, requires $O(N^3)$ to evalueate, so we consider an option to $M$ sub-samples and only consider the corresponding $M(M - 1)/2$ lines passing through every two of them. See pseudo-code in Algorithm 2.

For another option, we fix $K$ possible directions $(w_0, w_1) \in \mathbb{R}^2$, such that $w_0^2 + w_1^2 = 1$, and then for each of them consider the score $\hat{s}(X) = w_0 \hat{s}_0(X) + w_1 \hat{s}_1(X)$. This way we only need to choose the optimal thresholds $c$, similar to the case of DP. By doing so for each of the $K$ directions, we can approximate the best line in $O(K N_{val} \log N_{val})$ time (which includes sorting the score values). See detailed procedure in Algorithm 3.

# C   ABLATION STUDIES

Here we consider two ablation studies to examine the robustness of our method when the estimated conditional distributions are inaccurate. We demonstrate it with DP as fairness criterion, and we report the mean and standard error of the results from 3 independent runs.

---

**Algorithm 1** Algorithm for DP

---

**Input:** $\delta$: desired constraint for DP,
  $\{(X_i, Y_i, A_i)\}_{i=1}^{N_{val}}$: validation set,
  $\hat{p}(Y|X), \hat{p}(A|X)$,
  evaluated $\widehat{\Pr}(A=0), \widehat{\Pr}(A=1)$
**Output:** Threshold $t^*$ for modification rule $\kappa(X) = \mathbf{1}\{\hat{s}(X) > t^*\}$:
 1: Compute $\{\hat{Y}_i = \mathbf{1}\{\hat{p}(Y=1|X_i) > 0.5\}\}_{i=1}^{N_{val}}$
 2: $\texttt{acc} = \widehat{Acc}(\hat{Y})$
 3: $\texttt{counts}[a] = $ calculate frequency of $\hat{Y} = 1$ in two groups $A = 0, 1$
 4: Compute $\{\hat{s}(X_i)\}_{i=1}^{N_{val}}$ based on $\hat{p}(Y|X), \hat{p}(A|X)$, and $\widehat{Pr}(A)$
 5: Compute argsort $IDX = [i_1, \ldots, i_{N_{val}}]$ in ascending order, such that $\hat{s}(X_{i_j}) \geq \hat{s}(X_{i_{j+1}})$
 6: Best accuracy $ACC_{\max} = 0$
 7: **for** $j$ in $IDX$ **do**
 8:     set candidate $t = \hat{s}(X_j)$
 9:     $\texttt{acc} = \texttt{acc} + N_{val}^{-1}(1 - 2\mathbf{1}\{\hat{Y}_j = Y_j\})$               ▷ Update accuracy
10:     $\texttt{counts}[A_j] = \texttt{counts}[A_j] + N_{val}^{-1}(1 - 2\hat{Y}_j)$      ▷ Update counts for group $A = A_j$
11:     $\texttt{dp} = \left| \texttt{counts}[0]/\widehat{\Pr}(A=0) - \texttt{counts}[1]/\widehat{\Pr}(A=1) \right|$
12:     **if** $\texttt{dp} \leq \delta$ and $\texttt{acc} > ACC_{\max}$ **then**
13:         $t^* = t$                                      ▷ Update best candidate
14:         $ACC_{\max} = \texttt{acc}$
15:     **end if**
16: **end for**
17: **return** $t^*$

---

### C.1    CORRUPTED $\hat{p}(Y|X)$ AND $\hat{p}(A|X)$

In the first ablation study, we consider manually adding noise $\epsilon$ to $\hat{p}(Y|X)$ and $\hat{p}(A|X)$. For each $X$, we sample $\epsilon$ from a uniform distribution $Unif(-\alpha, 2\alpha)$, with $\alpha$ controlling the intensity of the corruption ($\alpha$ is chosen to be $0, 0.01, \ldots, 0.1$). In Figures 3 and 4, we plot the post-processed test target accuracy ($\check{Y}$) and post-processed test DP after selecting the modification threshold on validation set based on corrupted $\hat{p}(Y|X)$ and $\hat{p}(A|X)$ for Adult Census and CelebA (with target "Attractive" and sensitive attribute "Male") respectively. Across all corruption intensity $\alpha$, test DP still tends to be below or fluctuate around its desired constraints $\delta$'s, while we only incur a minor drop in test target accuracy (within 1% across most $\alpha$'s). This demonstrates the robustness of our post-processing algorithm even when the estimated conditional distributions do not match the ground-truth well.

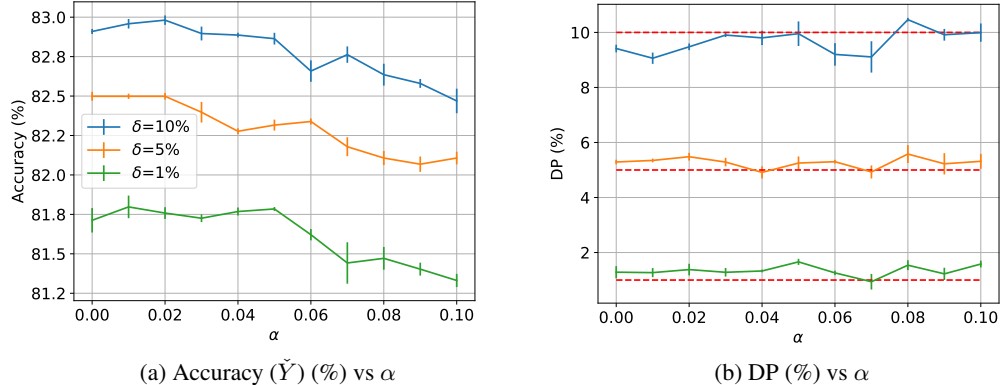

(a) Accuracy ($\check{Y}$) (%) vs $\alpha$                         (b) DP (%) vs $\alpha$

Figure 3: Test set performance of our method with corrupted $\hat{p}(Y|X)$ and $\hat{p}(A|X)$ on Adult Census dataset.

---

**Algorithm 2** Algorithm for EO

---

**Input:** $\delta$: desired constraint for EO,
  $M$: the number of random samples from the validation set
  $\{(X_i, Y_i, A_i)\}_{i=1}^{N_{val}}$: validation set,
  $\hat{p}(Y|X), \hat{p}(Y, A|X),$
  evaluated $\widehat{\Pr}(Y = y, A = a), a, y \in \{0, 1\}$
**Output:** Linear modification rule $\kappa(X)$

1: Compute $\{\hat{Y}_i = \mathbf{1}\{\hat{p}(Y = 1|X_i) > 0.5\}\}_{i=1}^{N_{val}}$
2: Compute $\{[\hat{s}_0(X_i), \hat{s}_1(X_i)]^\top\}_{i=1}^{N_{val}}$ based on $\hat{p}(Y|X), \hat{p}(Y, A|X)$ and $\widehat{\Pr}(Y, A)$
3: Randomly sample $M$ instances from the validation set: $\{(X'_m, Y'_m)\}_{m=1}^{M}$
4: Construct the set of linear rules to search with index:          $\triangleright$ size: $\frac{M(M-1)}{2}$
5:
$$\mathcal{S}_M = \{(k, l)|k < l, \ \ k, l = 1, \dots, M\}$$
6: Best accuracy $ACC_{\max} = 0$
7: **for** $(k, l)$ in $\mathcal{S}_M$ **do**
8:      $\kappa^{(k,l)}(X) = \mathbf{1}\left\{\frac{\hat{s}_1(X) - \hat{s}_1(X'_k)}{\hat{s}_1(X'_k) - \hat{s}_1(X'_l)} > \frac{\hat{s}_0(X) - \hat{s}_0(X'_k)}{\hat{s}_0(X'_k) - \hat{s}_0(X'_l)}\right\}$
9:      **for** $i = 1, \cdots, N_{val}$ **do**
10:         **if** $\kappa^{(k,l)}(X_i) = 1$ **then**
11:            $\check{Y}_i = 1 - \hat{Y}_i$                      $\triangleright$ Modify when $\kappa^{(k,l)}(X_i) = 1$
12:         **else**
13:            $\check{Y}_i = \hat{Y}_i$                        $\triangleright$ Do not modify when $\kappa^{(k,l)}(X_i) = 0$
14:         **end if**
15:      **end for**
16:      Compute validation $ACC(\kappa^{(k,l)})$ and $EO(\kappa^{(k,l)})$ based on $\check{Y}$
17:      **if** $EO(\kappa^{(k,l)}) \le \delta$ and $ACC(\kappa^{(k,l)}) > ACC_{max}$ **then**
18:         $\kappa(X) = \kappa^{(k,l)}(X), ACC_{max} = ACC(\kappa^{(k,l)})$
19:      **end if**
20: **end for**
21: **return** $\kappa(X)$

---

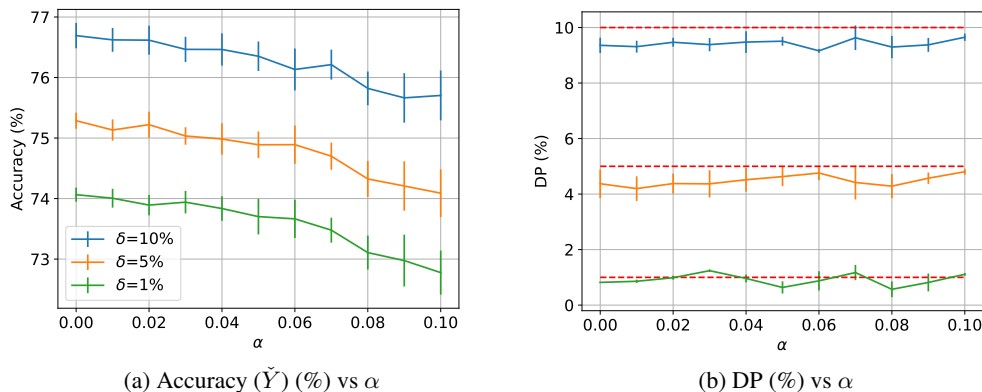

(a) Accuracy ($\check{Y}$) (%) vs $\alpha$                 (b) DP (%) vs $\alpha$

Figure 4: Test set performance of our method with corrupted $\hat{p}(Y|X)$ and $\hat{p}(A|X)$ on CelebA dataset (target: "Attractive", sensitive attribute: "Male").

## C.2 STRONGLY REGULARIZED $\hat{p}(A|X)$

In the second ablation study, we reduce the discrimination power of the sensitive classifier by training $\hat{p}(A|X)$ with stronger regularization. Specifically, for CelebA (target "Attractive" and sensitive attribute "Male"), we train multiple sensitive classifiers with different weight decay $\lambda$ increasing from 0.001 to 0.1. Figure 5 plots the test accuracy of sensitive model $\hat{p}(A|X)$, post-processed test

---

**Algorithm 3** Alternative algorithm for EO

---

**Input:** $\delta$: desired constraint for EO,
$\qquad \{(X_i, Y_i, A_i)\}_{i=1}^{N_{val}}$: validation set,
$\qquad \hat{p}(Y|X), \hat{p}(Y, A|X)$,
$\qquad$ evaluated $\widehat{\Pr}(Y = y, A = a), a, y \in \{0, 1\}$
**Output:** Threshold $t^*$ and direction $(w_0^*, w_1^*)$ for modification rule $\kappa(X) = \mathbf{1}\{w_0^* \hat{s}_0(X) + w_1^* \hat{s}_1(X) > t^*\}$:
 1: Compute $\{\hat{Y}_i = \mathbf{1}\{\hat{p}(Y = 1|X_i) > 0.5\}\}_{i=1}^{N_{val}}$
 2: $\texttt{counts}[y, a] =$ calculate frequency of $\hat{Y} = 1$ in all four groups $(Y, A) = (y, a)$
 3: Compute $\{(\hat{s}_0(X_i), \hat{s}_1(X_i))\}_{i=1}^{N_{val}}$ based on $\hat{p}(Y|X), \hat{p}(Y, A|X)$, and $\widehat{\Pr}(Y, A)$
 4: Fix set of directions $\mathcal{W}_K = \{(w_0^{(k)}, w_1^{(k)})\}_{k=1}^K \subset \mathbb{R}^2, \|(w_0^{(k)}, w_1^{(k)})\|_2 = 1$
 5: Best accuracy $ACC_{\max} = 0$
 6: **for** $(w_0, w_1)$ in $\mathcal{W}_K$ **do**
 7: $\qquad \texttt{acc} = \widehat{Acc}(\hat{Y})$
 8: $\qquad$ Compute scalar scores $\hat{s}(X_i) = w_0 \hat{s}_0(X_i) + w_1 \hat{s}_1(X_i)$
 9: $\qquad$ Compute argsort $IDX = \{i_1, \dots, i_{N_{val}}\}$ in ascending order, such that $\hat{s}(X_{i_j}) \geq \hat{s}(X_{i_{j+1}})$
10: $\qquad$ **for** $j$ in $IDX$ **do**
11: $\qquad\qquad$ set candidate $t = \hat{s}(X_j)$
12: $\qquad\qquad \texttt{acc} = \texttt{acc} + N_{val}^{-1}(1 - 2\mathbf{1}\{\hat{Y}_j = Y_j\})$ $\qquad\qquad\qquad\qquad \triangleright$ Update accuracy
13: $\qquad\qquad \texttt{count}[Y_j, A_j] = \texttt{count}[Y_j, A_j] + N_{val}^{-1}(1 - 2\hat{Y}_j)$ $\quad \triangleright$ Update counts for group $(Y_j, A_j)$
14: $\qquad\qquad \texttt{eo0} = \left| \texttt{count}[0, 0]/\widehat{\Pr}(Y = 0, A = 0) - \texttt{count}[0, 1]/\widehat{\Pr}(Y = 0, A = 1) \right|$
15: $\qquad\qquad \texttt{eo1} = \left| \texttt{count}[1, 0]/\widehat{\Pr}(Y = 1, A = 0) - \texttt{count}[1, 1]/\widehat{\Pr}(Y = 1, A = 1) \right|$
16: $\qquad\qquad \texttt{eo} = \max(\texttt{eo0}, \texttt{eo1})$
17: $\qquad\qquad$ **if** $\texttt{eo} \leq \delta$ and $\texttt{acc} > ACC_{\max}$ **then**
18: $\qquad\qquad\qquad ACC_{\max} = \texttt{acc}$ $\qquad\qquad\qquad\qquad\qquad\qquad\qquad \triangleright$ update best candidate
19: $\qquad\qquad\qquad t^* = t$
20: $\qquad\qquad\qquad (w_0^*, w_1^*) = (w_0, w_1)$
21: $\qquad\qquad$ **end if**
22: $\qquad$ **end for**
23: **end for**
24: **return** Modification rule $\kappa(X) = \mathbf{1}\{w_0^* \hat{s}_0(X) + w_1^* \hat{s}_1(X) > t^*\}$

---

target accuracy ($\check{Y}$), and post-processed DP after modification against $\lambda$ respectively. One can see that as $\lambda$ increases, the sensitive accuracy given by $\hat{p}(A|X)$ drops. However, with less accurate $\hat{p}(A|X)$, post-processed target accuracy ($\check{Y}$) and DP after modification does not seem to be noticeably hurt. Interestingly, when $\lambda$ increases moderately, although $\hat{p}(A|X)$ becomes less accurate, the post-processed target ($\check{Y}$) accuracy can even be improved. This further confirms the robustness of our post-processing modification algorithm.

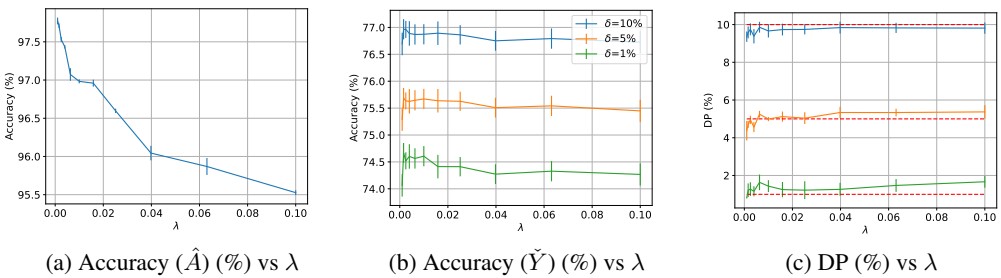

(a) Accuracy ($\hat{A}$) (%) vs $\lambda$ $\qquad$ (b) Accuracy ($\check{Y}$) (%) vs $\lambda$ $\qquad$ (c) DP (%) vs $\lambda$

Figure 5: Test set performance of our method based on $\hat{p}(A|X)$ trained with different weight decay $\lambda$ on CelebA dataset (target: "Attractive", sensitive attribute: "Male").

### C.3 ZERO-SHOT CLIP AS AUXILIARY MODEL

Table 2: Experiment with CLIP: Accuracy (%) and Equalized Odds (EO) (%) on CelebA with target attribute "Attractive" and sensitive attributes "Male". Main unconstrained model $\hat{p}(Y|X)$ is trained on CelebA, while auxiliary model $\hat{p}(Y, A|X)$ is evaluated from CLIP's similarity to text prompts.

|            | $\delta$ | ACC  | EO   |
|------------|----------|------|------|
|            | $\infty$ | 81.1 | 27.8 |
|            | 10       | 81.0 | 10.0 |
| Validation | 5        | 80.4 | 5.0  |
|            | 1        | 80.0 | 1.0  |
|            | $\infty$ | 81.6 | 24.1 |
|            | 10       | 80.5 | 6.2  |
| Test       | 5        | 79.9 | 1.0  |
|            | 1        | 79.1 | 5.4  |

Here, we consider for CelebA the classification problem with target "Attractive" and sensitive attribute "Male", with EO constraints imposed. We train only the target model $\hat{p}(Y|X)$ on the original CelebA data, while the pair $\hat{p}(Y, A|X)$ is evaluated from CLIP. Namely, we consider the following prompts,

$$\texttt{prompt\_11} = \text{``a photo of Attractive Male''}$$
$$\texttt{prompt\_10} = \text{``a photo of Attractive Female''}$$
$$\texttt{prompt\_01} = \text{``a photo of Unattractive Male''}$$
$$\texttt{prompt\_01} = \text{``a photo of Unattractive Female''}$$

For a given prompt and image $X$, let $sim(X, \texttt{prompt})$ denotes the CLIP's similarity. We consider the following probabilities,

$$\hat{p}(Y = i, A = j|X) = \frac{\exp(sim(X, \texttt{prompt\_}ij))}{\sum_{i,j \in \{0,1\}} \exp(sim(X, \texttt{prompt\_}ij)}$$

We then feed these two models into Algorithm 3. Results are reported in Table 2.

## D RESULTS IN TABLES

We present numerical results (mean±standard error) for all experiments on CelebA in tables.

Table 3: Accuracy (%) and Equalized Odds (EO) (%) on CelebA with target attribute "Attractive" and sensitive attributes "Male" under different desired levels ($\delta$'s) of EO (%) constraints. **Boldface** is used when MBS is better than Park et al. (2022) both in terms of accuracy and EO.

|  | $\delta$ | ACC | EO |
|---|---|---|---|
| | $\infty$ | 81.3±0.1 | 26.8±1.0 |
| | 10 | 80.8±0.1 | 9.9±0.0 |
| Validation | 5 | 80.2±0.2 | 5.0±0.0 |
| | 1 | 79.6±0.2 | 0.7±0.2 |
| | Hardt et al. (2016a) | 71.3±0.1 | 0.5±0.1 |
| | $\infty$ | 81.9±0.3 | 23.8±0.8 |
| | 10 | 80.9±0.2 | 6.7±0.6 |
| Test | 5 | **80.1±0.3** | **2.2±0.5** |
| | 1 | 79.3±0.3 | 4.0±0.4 |
| | Hardt et al. (2016a) | 71.7±0.3 | 3.8±0.2 |
| | Park et al. (2022) | 79.1±0.4 | 6.5±0.4 |

Table 4: Accuracy (%) and Equalized Odds (EO) (%) on CelebA with target attribute "Attractive" and sensitive attributes "Young" under different desired levels ($\delta$'s) of EO (%) constraints. **Boldface** is used when MBS is better than Park et al. (2022) both in terms of accuracy and EO.

|  | $\delta$ | ACC | EO |
|---|---|---|---|
| | $\infty$ | 81.3±0.0 | 27.7±0.4 |
| | 10 | 80.2±0.1 | 10.0±0.0 |
| Validation | 5 | 79.2±0.1 | 4.9±0.1 |
| | 1 | 78.2±0.2 | 1.0±0.0 |
| | Hardt et al. (2016a) | 70.4±0.1 | 0.4±0.2 |
| | $\infty$ | 82.0±0.0 | 24.7±0.8 |
| | 10 | **79.7±0.1** | **10.9±0.5** |
| Test | 5 | 78.2±0.0 | 6.6±0.8 |
| | 1 | 76.9±0.0 | 2.0±0.7 |
| | Hardt et al. (2016a) | 71.4±0.3 | 3.4±0.7 |
| | Park et al. (2022) | 79.1±0.5 | 12.4±0.5 |

Table 5: Accuracy (%) and Equalized Odds (EO) (%) on CelebA with target attribute "Big_Nose" and sensitive attributes "Male" under different desired levels ($\delta$'s) of EO (%) constraints. **Boldface** is used when MBS is better than Park et al. (2022) both in terms of accuracy and EO.

|  | $\delta$ | ACC | EO |
|---|---|---|---|
| | $\infty$ | 82.9±0.6 | 43.3±2.1 |
| | 10 | 82.7±0.1 | 9.9±0.1 |
| Validation | 5 | 82.3±0.1 | 5.0±0.0 |
| | 1 | 81.8±0.1 | 0.9±0.0 |
| | Hardt et al. (2016a) | 76.2±0.2 | 0.5±0.2 |
| | $\infty$ | 84.1±0.1 | 44.9±2.5 |
| | 10 | 83.5±0.2 | 8.4±1.2 |
| Test | 5 | **83.3±0.2** | **4.3±1.0** |
| | 1 | 83.2 ±0.1 | 1.7±0.3 |
| | Hardt et al. (2016a) | 78.2±0.0 | 0.5±0.0 |
| | Park et al. (2022) | 82.9±0.4 | 4.7±0.5 |

Table 6: Accuracy (%) and Equalized Odds (EO) (%) on CelebA with target attribute "Big_Nose" and sensitive attributes "Young" under different desired levels ($\delta$'s) of EO (%) constraints. **Boldface** is used when MBS is better than Park et al. (2022) both in terms of accuracy and EO.

|  | $\delta$ | ACC | EO |
|---|---|---|---|
| | $\infty$ | 83.5±0.0 | 21.5±0.1 |
| | 10 | 83.3±0.0 | 9.7±0.1 |
| Validation | 5 | 83.0±0.1 | 4.9±0.1 |
| | 1 | 82.2±0.2 | 1.0±0.0 |
| | Hardt et al. (2016a) | 79.0±0.0 | 0.6±0.1 |
| | $\infty$ | 84.5±0.1 | 21.0±0.1 |
| | 10 | 84.3±0.1 | 10.6±0.6 |
| Test | 5 | **84.2±0.1** | **4.1±0.8** |
| | 1 | 83.6±0.2 | 2.4±0.6 |
| | Hardt et al. (2016a) | 80.1±0.1 | 1.3±0.3 |
| | Park et al. (2022) | 84.1±0.5 | 4.8±0.3 |

Table 7: Accuracy (%) and Equalized Odds (EO) (%) on CelebA with target attribute "Bag_Under_Eyes" and sensitive attributes "Male" under different desired levels ($\delta$'s) of EO (%) constraints. **Boldface** is used when MBS is better than Park et al. (2022) both in terms of accuracy and EO.

|  | $\delta$ | ACC | EO |
|---|---|---|---|
| | $\infty$ | 84.5±0.0 | 19.8±3.8 |
| | 10 | 84.4±0.0 | 8.3±0.7 |
| Validation | 5 | 84.2±0.1 | 5.0±0.0 |
| | 1 | 82.3±0.1 | 1.0±0.0 |
| | Hardt et al. (2016a) | 79.8±0.2 | 0.4±0.1 |
| | $\infty$ | 85.2±0.1 | 19.0±3.7 |
| | 10 | 85.0±0.0 | 8.2±0.7 |
| Test | 5 | 84.7±0.1 | 5.6±0.2 |
| | 1 | 83.1±0.0 | 1.7±0.2 |
| | Hardt et al. (2016a) | 80.2±0.2 | 1.7±0.4 |
| | Park et al. (2022) | 83.4±0.6 | 3.0±0.4 |

Table 8: Accuracy (%) and Equalized Odds (EO) (%) on CelebA with target attribute "Bag_Under_Eyes" and sensitive attributes "Young" under different desired levels ($\delta$'s) of EO (%) constraints. **Boldface** is used when MBS is better than Park et al. (2022) both in terms of accuracy and EO.

|  | $\delta$ | ACC | EO |
|---|---|---|---|
| | $\infty$ | 84.6±0.1 | 18.2±1.1 |
| | 10 | 84.6±0.0 | 9.9±0.0 |
| Validation | 5 | 84.4±0.0 | 5.0±0.0 |
| | 1 | 83.0±0.1 | 1.0±0.0 |
| | Hardt et al. (2016a) | 81.1±0.2 | 0.5±0.2 |
| | $\infty$ | 85.6±0.3 | 13.7±1.9 |
| | 10 | 85.1±0.1 | 9.5±1.0 |
| Test | 5 | 85.1±0.0 | 5.4±0.5 |
| | 1 | 83.6±0.2 | 1.8±0.3 |
| | Hardt et al. (2016a) | 81.7±0.2 | 1.6±0.5 |
| | Park et al. (2022) | 83.5±0.3 | 1.6±0.3 |

# E  SENSITIVITY ANALYSIS FOR COMPOSITE CRITERION

For the sake of convenience, we assume that the classes are valued as $\{-1, +1\}$ instead of $0, 1$. In such a case, the unconstrained optimal classifier that has the form $Y^*(X) = \text{sign}(\hat{p}(Y = 1|X) - 0.5)$ and constrained optimal classifiers have form $Y^*_{\boldsymbol{\gamma}}(X) = Y^*(X) \, \text{sign}\left(\sum_j \gamma_j s_j(X) - 1\right)$, i.e. ones that have highest accuracy subject to restriction $CC \leq \delta$.

In the MBS implementation (Section 3), we replace the unknown ground-truth probabilities $p(Y|X)$ and $p(A|X)$ ($p(Y, A|X)$ for EO) with their estimations, which we denote as $\hat{p}(Y|X)$ and $\hat{p}(A|X)$ ($\hat{p}(Y, A|X)$), respectively. The empirical version of the unconstrained classifier has the form $\hat{Y}(X) = \text{sign}(\hat{p}(Y = 1|X) - 0.5)$ and we are looking for a classifier of the form $\check{Y}_{\boldsymbol{\gamma}}(X) = \hat{Y}(X) \, \text{sign}\{\sum_j \gamma_j \hat{s}_j(X) - 1\}$, i.e. our goal is to fit a set of $k$ parameters $\boldsymbol{\gamma} = (\gamma_j) \in \mathbb{R}^k$. Recall that the composite criterion has the form Eq. 4, and the score functions are defined in Eq. 8, Eq. 7.

We assume that we choose the score weights $\boldsymbol{\gamma} = (\gamma_1, \ldots, \gamma_k)$ based on the observed validation sample $\{(X_i, Y_i, A_{1i}, A_{2i}, \ldots, A_{ki})\}_{i=1}^N$ of size $N$ by maximizing the corresponding empirical accuracy, while maintaining the restriction on empirical parity. Set,

$$\widehat{Acc}(\check{Y}) = \frac{1}{N} \sum_{i=1}^N \mathbf{1}\{Y_i = \check{Y}(X_i)\},$$

$$\widehat{CC}(\check{Y}) = \max_{j \leq k} |\widehat{C}_j(\check{Y})|$$

$$\widehat{C}_j(\check{Y}) = \frac{1}{N} \sum_{i=1}^N \mathbf{1}\{\check{Y}(X_i) = +1\} \left[ \frac{\mathbf{1}\{A_{ji} = a_j\}}{\Pr(A_{ji} = a_j)} - \frac{\mathbf{1}\{A_{ji} = b_j\}}{\Pr(A_{ji} = b_j)} \right].$$

For a given level $\delta > 0$ we fit the parameters on validation as follows

$$\hat{\boldsymbol{\gamma}} = \arg\max\{\widehat{Acc}(\check{Y}_{\boldsymbol{\gamma}}) : \widehat{DP}(\check{Y}_{\boldsymbol{\gamma}}) \leq \delta\}. \tag{17}$$

The goal of this section is to demonstrate that, if we are provided an accurate enough estimate $\hat{p}$ and a sufficiently large validation set $N$, we can have guaranties for the empirically derived estimator $\check{Y}_{\hat{\boldsymbol{\gamma}}}(X)$. We deliberately do not touch the question of what the estimation error of $p(Y, A|X)$ could be, which may depend on various factors such as training sample size, input dimensionality, and complexity of the parametrized model (Vapnik, 1998; Boucheron et al., 2005), as well as the training algorithm (Bousquet & Elisseeff, 2002; Hardt et al., 2016b; Klochkov & Zhivotovskiy, 2021).

**Theorem 2.** *Assume the following set of conditions:*

1. *We have estimation of conditional probabilities that satisfy moment bounds (for $p \geq 1$),*

$$\mathbb{E}^{1/p}|\hat{p}(Y = 1|X) - p(Y = 1|X)|^p \leq \varepsilon_p,$$
$$\mathbb{E}^{1/p}|\hat{p}(A_j = a_j|X) - p(A_j = b_j|X)|^p \leq \varepsilon_p, \qquad \forall j = 1, \ldots, k,$$
$$\mathbb{E}^{1/p}|\hat{p}(A_j = b_j|X) - p(A_j = b_j|X)|^p \leq \varepsilon_p, \qquad \forall j = 1, \ldots, k;$$

2. *The distribution satisfies the* margin assumption $\eta(X) > \eta^*$ *for all $X$ (known as Mammen-Tsybakov condition (Mammen & Tsybakov, 1999));*

3. *All groups have significant size: we have $\Pr(A_j = a_j), \Pr(A_j = b_j) \geq p_0$;*

4. *Score distribution regularity: the distribution of scores $\mathbf{s}(X) = (s_1(X), \ldots, s_k(X))$ is supported on a ball $\{s \in \mathbb{R}^k : \|s\| \leq B\}$; Moreover, for every $\|\boldsymbol{\gamma}\| = 1$, the density of $\boldsymbol{\gamma}^\top \mathbf{s}(X)$ is bounded by a constant $L$;*

5. *The weights corresponding to optimal $Y^*_{\boldsymbol{\gamma}}$ under $CC(Y^*_{\boldsymbol{\gamma}}) \leq \delta/2$ satisfy $\|\boldsymbol{\gamma}\|_1 \leq R$;*

*There is a constant $C = C(\eta^*, p_0, L, R, B, \delta)$, such that, for any $r \in (0, 1)$ that satisfies,*

$$C\left( \sqrt{\frac{k \log(kN/r)}{N}} + (\sqrt{k}\varepsilon_p)^{1 - \frac{1}{1+p}} \right) \leq 1, \tag{18}$$

*we have that, with probability at least $1 - r$,*

$$Acc(\check{Y}_{\hat{\gamma}}) \geq Acc^*(\delta) - C\left((\sqrt{k}\varepsilon_p)^{1-\frac{1}{p+1}} + \sqrt{\frac{k\log(kN/r)}{N}}\right),$$

$$CC(\check{Y}_{\hat{\gamma}}) \leq \delta + C\sqrt{\frac{k\log(kN/r)}{N}},$$

$$(19)$$

*where $Acc^*(\delta)$ is the accuracy of the optimal classifier under the constraints $CC \leq \delta$.*

Before we move on to the proof, let us briefly comment on each condition. The first condition requires that we have a good estimation of probabilities in the form of moment bound. We note that bounds of this form often appear in theoretical analysis of non-parametric regression Tsybakov (2009), including ones based on neural networks Hu et al. (2021). Although they are standard in non-parametric regression, there is no way to verify them in practice without making structural assumption about the underlying distribution. The margin condition is very popular in Statistical Learning Theory, and in our particular case, it allows controlling the accuracy of the estimator under perturbations of the conditionals. We note that Denis et al. (2021) also assume a form of margin condition for the multi-class fair classification. The third condition is somewhat realistic and simply makes sure we have an appropriate criterion evaluation based on the validation sample. The fourth condition is a stronger version of the condition in Theorem 1, where we require that the distribution of $\boldsymbol{\gamma}^\top \mathbf{s}(X)$ is continuous. We prefer to have a much stronger version to avoid technical difficulties in the proof. As for the fifth condition, it is pretty much guaranteed that the vectors $\boldsymbol{\gamma}_\delta$ are bounded, which we show for instance as part of the proof (see Eq. 16 and the line below it). However, it is hard to precisely characterize the value of $R$ based on characteristic of this distribution, so we prefer to state it as a condition.

Furthermore we note that practical implementations based on Bayes optimal classifiers are often based on closed form derivations of the optimal weights Denis et al. (2021); Zeng et al. (2022). We do not provide these, instead our method treats the bias scores as features when fitting the modification rule. As a result, the error of estimation of the conditionals does not propagate as much into the fairness constraint. See also some empirical studies in Appendix C.

*Proof.* We compare the accuracy and composite criterion of $Y_{\boldsymbol{\gamma}}^*$ and $\check{Y}_{\hat{\gamma}}$ in three steps:

1. *Approximation on population level*: we bound the discrepancy in the population statistics ($Acc$ and $CC$) of estimators $Y_{\boldsymbol{\gamma}}^*$ and $\check{Y}_{\boldsymbol{\gamma}}$, i.e. how the difference between conditionals $p(Y|X), p(A_j|X)$ and its estimation propagates into the accuracy and composite criterion.

2. *Approximation on validation set*: we then compare the population statistics and validation statistics ($\widehat{Acc}, \widehat{CC}$) simultaneously for all estimators $Y_{\boldsymbol{\gamma}}^*, \check{Y}_{\boldsymbol{\gamma}}$.

3. *Connecting validation and population*: once the first two steps are done, we use the standard trick in empirical process theory to connect the classifiers that are optimal in population and in validation.

**Step 1 (approximation on population level).** We first check how the population accuracy and DP compare between the optimal $Y^*(X)$ and the one based on the estimated probabilities $\hat{Y}(X)$.

For the unconstrained classifiers, thanks to the margin condition, we have by Markov inequality:

$$\Pr(\hat{Y} \neq Y^*) = \Pr(\hat{p}(Y|X) < 0.5 < p(Y|X) \text{ or } p(Y|X) < 0.5 < \hat{p}(Y|X))$$
$$\leq \Pr(|\hat{p}(Y|X) - p(Y|X)| > \eta^*)$$
$$\leq \left(\frac{\varepsilon_p}{\eta^*}\right)^p.$$

In particular, this implies that $Acc(\hat{Y}) \geq Acc(Y^*) - (\varepsilon_p/\eta^*)^p$. In order to compare the accuracies of constrained classifiers, we need to further compare the modification rules. Denote $\kappa_{\boldsymbol{\gamma}}(X) = \text{sign}\{\boldsymbol{\gamma}^\top \mathbf{s}(X) - 1\}$ for the ground truth score, and consider empirically derived rule $\hat{\kappa}_{\boldsymbol{\gamma}}(X) =$

$\text{sign}\{\boldsymbol{\gamma}^\top \hat{\mathbf{s}}(X) - 1\}$, where $\hat{\mathbf{s}}(X)$ are derived by replacing $p(A_j = a_j|X)$ and $p(A_j = b_j|X)$ with $\hat{p}(A_j = a_j|X)$ and $\hat{p}(A_j = b_j|X)$, respectively. Let also $\hat{f}_j(X)$, $\hat{\eta}(X)$ be the corresponding substitutes for $f_j(X), \eta(X)$. We have,

$$
\begin{aligned}
\Pr(\kappa_{\boldsymbol{\gamma}} < \hat{\kappa}_{\boldsymbol{\gamma}}) &= \Pr(\boldsymbol{\gamma}^\top \mathbf{s}(X) < 1 < \boldsymbol{\gamma}^\top \hat{\mathbf{s}}(X)) \\
&= \Pr\left(\boldsymbol{\gamma}^\top \mathbf{s}(X) \in \left[1 + \boldsymbol{\gamma}^\top(\mathbf{s}(X) - \hat{\mathbf{s}}(X)); 1\right]\right) \\
&= \Pr\left(\tilde{\boldsymbol{\gamma}}^\top \mathbf{s}(X) \in \left[1/\|\boldsymbol{\gamma}\| + \tilde{\boldsymbol{\gamma}}^\top(\mathbf{s}(X) - \hat{\mathbf{s}}(X)); 1/\|\boldsymbol{\gamma}\|\right]\right),
\end{aligned}
$$

where we denote the normalized weights $\tilde{\boldsymbol{\gamma}} = \boldsymbol{\gamma}/\|\boldsymbol{\gamma}\|$. Taking the union with the opposite inequality, we get that

$$
\Pr(\kappa_{\boldsymbol{\gamma}} \neq \hat{\kappa}_{\boldsymbol{\gamma}}) \leq \Pr\left(\tilde{\boldsymbol{\gamma}}^\top \mathbf{s}(X) \in \left[1/\|\boldsymbol{\gamma}\| - |\tilde{\boldsymbol{\gamma}}^\top(\mathbf{s}(X) - \hat{\mathbf{s}}(X))|; 1/\|\boldsymbol{\gamma}\| + |\tilde{\boldsymbol{\gamma}}^\top(\mathbf{s}(X) - \hat{\mathbf{s}}(X))|\right]\right). \tag{20}
$$

To proceed, we want to bound the value $\tilde{\boldsymbol{\gamma}}^\top(\mathbf{s}(X) - \hat{\mathbf{s}}(X))$. Firstly, it is straightforward to see that $\mathbb{E}^{1/p}|\eta - \hat{\eta}|^p \leq 2\varepsilon_p$. For the $f$-score, we see that

$$
|f_j - \hat{f}_j| \leq \frac{1}{p_0}\mathbf{1}\{\hat{Y} \neq Y^*\} + \frac{1}{p_0}|p(A_j = a_j|X) - \hat{p}(A_j = a_j|X)| + \frac{1}{p_0}|p(A_j = b_j|X) - \hat{p}(A_j = b_j|X)|,
$$

so that by the triangle inequality,

$$
\mathbb{E}^{1/p}|f - \hat{f}|^p \leq p_0^{-1}\left(\frac{\varepsilon_p}{\eta^*}\right) + 2p_0^{-1}\varepsilon_p \leq \frac{3\varepsilon_p}{p_0\eta^*}.
$$

Then, we can write

$$
\begin{aligned}
\mathbb{E}^{1/p}|\hat{\eta}\tilde{\boldsymbol{\gamma}}^\top(s(X) - \hat{s}(X))|^p &\leq \mathbb{E}^{1/p}\left|\frac{\hat{\eta}}{\eta}\sum_j(f_j - \hat{f}_j)\tilde{\gamma}_j\right|^p + \mathbb{E}^{1/p}\left|\left(\frac{\hat{\eta}}{\eta} - 1\right)\sum_j \tilde{\gamma}_j \hat{f}_j\right|^p \\
&\leq \frac{1}{\eta^*}\mathbb{E}^{1/p}\left|\sum_j|f_j - \hat{f}_j|^2\right|^{p/2} + \frac{2\sqrt{k}}{p_0\eta^*}\mathbb{E}^{1/p}|\eta - \hat{\eta}|^p \\
&\leq \frac{3\sqrt{k}}{p_0\eta^*}\varepsilon_p.
\end{aligned}
$$

Using the Markov inequality, this gives us the bound for any $r \in (0, 1)$,

$$
\Pr\left(|\hat{\eta}\tilde{\boldsymbol{\gamma}}^\top(s(X) - \hat{s}(X))| \leq r^{-1}\frac{5\sqrt{k}}{p_0\eta^*}\varepsilon_p\right) \geq 1 - r^p.
$$

Using the moment bound on $\eta - \hat{\eta}$, we can also lowerbound $\hat{\eta}$,

$$
\Pr(\hat{\eta} \geq \eta^*/2) \leq \Pr(|\hat{\eta} - \eta| \leq \eta^*/2) \leq 1 - \left(\frac{4\varepsilon_p}{\eta^*}\right)^p.
$$

Taking the union bound of the two last displays, we get that

$$
\Pr\left(|\tilde{\boldsymbol{\gamma}}^\top(s(X) - \hat{s}(X))| \leq r^{-1}\frac{6\sqrt{k}}{p_0(\eta^*)^2}\varepsilon_p\right) \leq 1 - \left(\frac{4\varepsilon_p}{\eta^*}\right)^p - r^p.
$$

Now, we can plug this back into Eq. 20. We get that,

$$
\Pr(\kappa_{\boldsymbol{\gamma}} \neq \hat{\kappa}_{\boldsymbol{\gamma}}) \leq 2Lr^{-1}\frac{6\sqrt{k}}{p_0(\eta^*)^2}\varepsilon_p + r^p + \left(\frac{4\varepsilon_p}{\eta^*}\right)^p.
$$

Optimizing $r = (12L\sqrt{k}\varepsilon_p/(pp_0(\eta^*)^2))^{\frac{1}{p+1}}$, we get that

$$
\Pr(\kappa_{\boldsymbol{\gamma}} \neq \hat{\kappa}_{\boldsymbol{\gamma}}) \leq 24\left(\frac{L\sqrt{k}\varepsilon_p}{p_0(\eta^*)^2}\right)^{1-1/(p+1)} + \left(\frac{4\varepsilon_p}{\eta^*}\right)^p.
$$

We now can derive using the triangle inequality,

$$
\begin{aligned}
|Acc(\check{Y}_{\boldsymbol{\gamma}}) - Acc(Y_{\boldsymbol{\gamma}}^*)| &\leq \Pr(\check{Y}_{\boldsymbol{\gamma}} \neq Y_{\boldsymbol{\gamma}}^*) \\
&\leq \Pr(\hat{Y} \neq Y^*) + \Pr(\kappa_{\boldsymbol{\gamma}} \neq \hat{\kappa}_{\boldsymbol{\gamma}}) \\
&\leq 2\left(\frac{4\varepsilon_p}{\eta^*}\right)^p + 24\left(\frac{L\sqrt{k}\varepsilon_p}{p_0(\eta^*)^2}\right)^{1-1/(p+1)} \\
&\leq L_1(\sqrt{k}\varepsilon_p)^{1-1/(p+1)},
\end{aligned}
\tag{21}
$$

where we assume that $\varepsilon_p < \eta^*/4$ (condition Eq. 18) and set $L_1 = 4/\eta^* + 24\max((L/(p_0\eta^*))^{1-1/(p+1)}, 1)$. Similarly,

$$
|CC(\check{Y}_{\boldsymbol{\gamma}}) - CC(Y_{\boldsymbol{\gamma}}^*)| \leq \max_j |C_j(\check{Y}_{\boldsymbol{\gamma}}) - C_j(Y_{\boldsymbol{\gamma}}^*)| \leq p_0^{-1}\Pr(\check{Y}_{\boldsymbol{\gamma}} \neq Y_{\boldsymbol{\gamma}}^*) \leq L_2(\sqrt{k}\varepsilon_p)^{1-1/(p+1)},
$$

where $L_2 = p_0^{-1}L_1$.

**Step 2 (approximation on validation set).** The next step is to derive uniform bounds for the empirical accuracy and CC. We have,

$$
\widehat{Acc}(\check{Y}_{\boldsymbol{\gamma}}) = \frac{1}{2N}\sum_{i=1}^{N} \hat{Y}(X_i)Y_i \operatorname{sign}(\boldsymbol{\gamma}^\top \hat{\mathbf{s}}(X_i) - 1) + \frac{1}{2}
$$

$$
\widehat{Acc}(Y_{\boldsymbol{\gamma}}^*) = \frac{1}{2N}\sum_{i=1}^{N} Y^*(X_i)Y_i \operatorname{sign}(\boldsymbol{\gamma}^\top \mathbf{s}(X_i) - 1) + \frac{1}{2}
$$

and notice that $\mathbb{E}_{val}\widehat{Acc}(\check{Y}_{\boldsymbol{\gamma}}) = Acc(\check{Y}_{\boldsymbol{\gamma}})$ and $\mathbb{E}_{val}\widehat{Acc}(Y_{\boldsymbol{\gamma}}^*) = Acc(Y_{\boldsymbol{\gamma}}^*)$, where we conventionally denote $\mathbb{E}_{val}$ as the expectation (and below $\Pr_{val}$ for probability) w.r.t. the validation sample.

**Lemma 2** (Vapnik (1998), Theorem 5.3). *Suppose, we have a bounded function $f(Z) \in [-1,1]$, and $k$ arbitrary functions $g_1(X), g_2(X), \ldots, g_k(X)$, and consider a class of functions $\{\lambda_{\boldsymbol{\gamma}}(X) = f(Z)\operatorname{sign}\{\sum_j \gamma_j g_j(Z) - 1\} : \boldsymbol{\gamma} \in \mathbb{R}^k\}$. Let $X_1, \ldots, X_N$ be i.i.d. Then, we have with probability $1 - \delta$ that for each $\boldsymbol{\gamma}$,*

$$
\left|\frac{1}{N}\sum_i \lambda_{\boldsymbol{\gamma}}(Z_i) - \mathbb{E}\lambda_{\boldsymbol{\gamma}}(Z)\right| \leq L_0\sqrt{\frac{k\log(N/\delta)}{N}}.
$$

*Proof.* We simply need to observe that the shatter coefficient of set $\{\lambda_{\boldsymbol{\gamma}}(X) < \tau\}$ is bounded by $(N+1)^{k+1}$, then apply Theorem 5.3 from Vapnik (1998). $\qquad \square$

For $Z = (X, Y)$ set $f(Z) = \hat{Y}(X)Y/2$, $g_j(Z) = \hat{s}_j(X)$ to control the accuracy of $\check{Y}_{\boldsymbol{\gamma}}(X)$ and $f(Z) = Y^*(X)Y/2$, $g_j(Z) = s_j(X)$, to control the accuracy of $Y_{\boldsymbol{\gamma}}^*(X)$. Furthermore, for $Z = (X, Y, A_1, \ldots, A_k)$ we can set $f(Z) = \hat{Y}(X)\left[\Pr(A_j = a_j)^{-1}\mathbf{1}\{A_j = a_j\} - \Pr(A_j = b_j)^{-1}\mathbf{1}\{A_j = b_j\})\right]/2$, $g(Z) = \operatorname{sign}(\boldsymbol{\gamma}^\top \hat{\mathbf{s}}(X) - 1)$ to get concentration of disparity $C_j$ for $\check{Y}_{\boldsymbol{\gamma}}(X)$, and similarly for $Y_{\boldsymbol{\gamma}}^*(X)$. Overall, we have that for any $r \in (0, 1)$ each of the bounds holds

$$
\left|\widehat{Acc}(\check{Y}_{\boldsymbol{\gamma}}) - Acc(\check{Y}_{\boldsymbol{\gamma}})\right| \leq L_0\sqrt{\frac{k\log(N/r)}{N}} \qquad \text{w. p.} \qquad \geq 1 - r
$$

$$
\left|\widehat{Acc}(Y_{\boldsymbol{\gamma}}^*) - Acc(Y_{\boldsymbol{\gamma}}^*)\right| \leq L_0\sqrt{\frac{k\log(N/r)}{N}} \qquad \text{w. p.} \qquad \geq 1 - r
$$

$$
\left|\widehat{C}_j(\check{Y}_{\boldsymbol{\gamma}}) - C_j(\check{Y}_{\boldsymbol{\gamma}})\right| \leq L_0 p_0^{-1}\sqrt{\frac{k\log(N/r)}{N}} \qquad \text{w. p.} \qquad \geq 1 - r
$$

$$
\left|\widehat{C}_j(Y_{\boldsymbol{\gamma}}^*) - C_j(Y_{\boldsymbol{\gamma}}^*)\right| \leq L_0 p_0^{-1}\sqrt{\frac{k\log(N/r)}{N}} \qquad \text{w. p.} \qquad \geq 1 - r
$$

If we take $r = r/k$ for the bound on each $C_j$ and then take a union bound, we obtain a bound on the difference in the composite criterion. Overall, we have that with probability at least $1 - r$,

$$\left|\widehat{Acc}(\check{Y}_{\gamma}) - Acc(\check{Y}_{\gamma})\right| \le L_3 \sqrt{\frac{k \log(N/r)}{N}}$$

$$\left|\widehat{Acc}(Y_{\gamma}^*) - Acc(Y_{\gamma}^*)\right| \le L_3 \sqrt{\frac{k \log(N/r)}{N}}$$

$$\left|\widehat{CC}(\check{Y}_{\gamma}) - CC(\check{Y}_{\gamma})\right| \le L_3 \sqrt{\frac{k \log(kN/r)}{N}}$$

$$\left|\widehat{CC}(Y_{\gamma}^*) - CC(Y_{\gamma}^*)\right| \le L_3 \sqrt{\frac{k \log(kN/r)}{N}} .$$

where we set $L_3 = L_0 p_0^{-1} \sqrt{\log 4}$.

**Step 3 (connecting validation and population).** Let $\gamma$ corresponds to optimal $Y_{\gamma}^*$ under $CC(Y_{\gamma}^*) \le \delta$ and denote its accuracy $Acc^*(\delta)$. Then, let $\gamma'$ corresponds to the the optimal $\check{Acc}(Y_{\gamma'}^*)$ under the constraint $CC(Y_{\gamma'}^*) \le \delta - \epsilon$. At the end of this section we show that

$$Acc(Y_{\gamma'}^*) = Acc^*(\delta - \epsilon) \ge Acc^*(\delta) - \epsilon R \quad \text{for} \quad \epsilon < \delta/2, \tag{22}$$

and let us assume for now that it is true. Set $\epsilon_1 = L_3 \sqrt{\frac{k \log(kN/r)}{N}}, \epsilon_2 = L_2 \varepsilon_p^{1-1/(p+1)}, \epsilon = \epsilon_1 + \epsilon_2$, and note that by assumption Eq. 18, $\epsilon < \delta/2$. Then, $\widehat{CC}(\check{Y}_{\gamma'}) \le CC(\check{Y}_{\gamma'}) + L_3 \sqrt{\frac{k \log(kN/\gamma)}{N}} \le CC(\check{Y}_{\gamma'}^*) + \epsilon = \delta$, which means that the problem Eq. 17 is feasible, and also $\widehat{Acc}(\check{Y}_{\hat{\gamma}}) \ge \widehat{Acc}(\check{Y}_{\gamma'}) \ge Acc(\check{Y}_{\gamma'}) - \epsilon_1 \overset{Eq.\ 21}{\ge} Acc(Y_{\gamma'}^*) - \epsilon_2 - \epsilon_1 = Acc^*(\delta - \epsilon) - \epsilon$. Finally, it implies that

$$Acc(\check{Y}_{\hat{\gamma}}) \ge Acc^*(\delta) - (2 + R)\epsilon$$
$$CC(\check{Y}_{\hat{\gamma}}) \le \delta - \epsilon_1.$$

Substituting $\epsilon, \epsilon_1$, we finally get the bound stated in Eq. 19.

**Check of Eq. 22.** Let us denote $\gamma_{\delta}$ to be (some) optimal set of weights, corresponding to the classifier that maximizes the accuracy under restriction $CC(Y_{\gamma}^*) \le \delta$. We show that for $\epsilon < \delta$,

$$Acc(\delta - \epsilon) - Acc(\delta) \le \epsilon \|\gamma_{\delta-\epsilon}\|_1,$$

which thanks to condition 5. yields Eq. 22. To show that, we need to recall the proof of Theorem 1, where $\gamma_{\delta}$ comes from the dual LP problem (Lemma 1), and the value $Acc^*(\delta)$ corresponds to the maximal objective in the primal and minimal objective in the dual LP. Changing $\delta \mapsto \delta - \epsilon$ simply means that we change the vector $b$ in Eq. 15 to a new $b^{new} = b - \varepsilon \begin{pmatrix} \mathbf{1}_k \\ -\mathbf{1}_k \end{pmatrix}$, where $\mathbf{1}_k$ denotes a column with $k$ ones. Let $z^{new}, \lambda^{new}$ denotes the solution to dual LP with $b$ replaced by $b^{new}$ in Eq. 15. Then, since in the dual LP the constraints do not depend on $b$, the value at $z^{new}, \lambda^{new}$ serves as an upperbound for the old problem. Therefore, $Acc(\delta) - Acc(\delta - \varepsilon) \le z^{new}(b^{new} - b) \le \varepsilon \|\gamma_{\delta-\varepsilon}\|_1$, since the corresponding $\gamma_{\delta-\varepsilon}$ has the coordinates $z_j - z_{j+k}$. $\qquad \square$

# F  COMPARISON WITH REDUCTION METHOD (AGARWAL ET AL., 2018)

Here, we include Reduction method (Agarwal et al., 2018) as an additional baseline, and we use the same network architecture and hyperparameters as for the other methods.

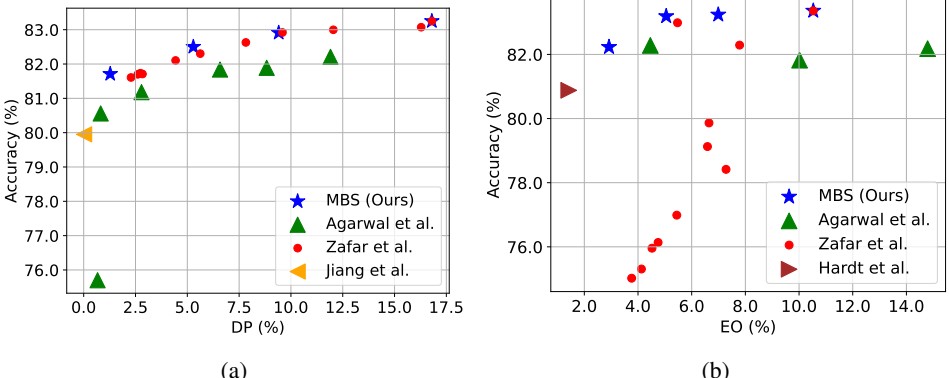

(a)

(b)

Figure 6: Accuracy (%) vs (a) Demographic Parity (DP) (%) and (b) Equalized Odds (EO) trade-offs on Adult Census. Desired $\delta = \infty$ (unconstrained), $10\%$, $5\%$, and $1\%$.

