# OpenReview forum: "Post-hoc bias scoring is optimal for fair classification"
_ICLR.cc/2024/Conference — ICLR 2024 spotlight_

### Official Review · Reviewer_v53s · 2023-10-24

**Soundness:** 2 fair
**Presentation:** 3 good
**Contribution:** 3 good
**Rating:** 6
**Confidence:** 4

**Summary:**

This paper proposes a post-processing method for achieving fairness on binary classification problems, which leverages a representation result for the Bayes optimal fair classifier as a (linear) function of the bias scores, which are given by a function of several conditional probabilities.  The results cover the fairness criteria of DP, EO and EOp, can handle multiple sensitive attributes, and most notably, is applicable to the attribute-unaware setting (i.e., sensitive attribute is not observed during inference).

---

Post-rebuttal: I have increased my score, although I feel the assumptions for theorem 2 are unnecessarily strong/complicated.

**Strengths:**

- As mentioned in the summary, the results are general in that they cover DP, EO and EOp, can handle multiple sensitive attributes, and most notably, is applicable to the attribute-unaware setting (i.e., sensitive attribute is not observed during inference).
- The framework is flexible in allowing for composite criteria.
- The authors provides some qualitative interpretation of the representation result (theorem 1), namely the bias score, which practitioners may find helpful.
- Paper is well-written, and the main body is mostly easy-to-follow.

All in all, I like the representation result, which I think is neat in that it encompasses many learning settings, but my rating is limited by my opinion that the current version of the manuscript is incomplete, as detailed in the weaknesses.

**Weaknesses:**

1. It is not mentioned how nonbinary sensitive attributes are handled.

	- Related: How is the sensitive attribute of "race" in the COMPAS experiments, which can be one of three categories (African-American, Caucasian, Other), handled?

1. I am skeptical about the scalability of the proposed method to large datasets and more sensitive attributes.  It appears that the time complexity would scale exponentially in the number of sensitive attributes, i.e., $M^{K}$.

	- I see that $M$, the number of samples on which the decision boundaries are considered, is set to no more than 5000 in the experiments.  But practical ML datasets nowadays can contain tens of thousands to millions of examples.
	- Could the authors also report and compare the running time of their code?
	- Also, if $M<N_\textrm{val}$ is used for selecting the boundaries, then there should be a term involving $M$ in Theorem 2 (more on Theorem 2 below)?

1. Theorem 2 is very important as it provides fairness guarantees for the classifier obtained through the procedure.  But the result looks wrong to me, and seems to have a discrepancy with Algorithm 1.  The proof is also very hard to read, containing several typos.

	- DP should depend on $\epsilon_p$ (attributed to the error in $\hat p(A=1\mid X$), but this dependency is absent in eq. 14.  Digging into the proof, I see that it is hidden with the statement that "$\epsilon=\epsilon_1+\epsilon_2$, and assume that it is smaller than $\delta/2$".  Why is this assumption justified?
	- In the paragraph preceding eq. 15, what is $\hat Y_{t'}$?  Should it be $\check Y_{t'}$?  And what is $\check Y_{t'}^*$ in the paragraph following eq. 15?  Should it be $\check Y_{t'}$?
	- Following the above, I don't get why $DP(\check Y_{t'})+\epsilon \leq \delta$ but not $3\delta/2$, given that "let $t'$ corresponds to... under the constraint that $DP(\check Y_{t'})\leq \delta-\epsilon$".
	- I don't get why $DP(\check Y_{\hat t})\leq \delta -\epsilon_1$, how is $\check Y_{\hat t}$ related to $\check Y_{t'}$?
	- Finally, Theorem 2 only proves results for DP.  What about EO, EOp, and composite criteria?
	- Please also justify the assumptions made; are they practical?

1. In the experiments, the authors compared their proposed post-processing algorithm to ones that are attribute-aware, but their algorithm is run in attribute-unaware mode.  The authors should have compared to those algorithms by running their algorithm in the same attribute-aware mode.  In this sense, the current set of experiments is incomplete.

1. The conclusions drawn from the ablation study in section D.2 do no make sense to me.  How is accuracy related to the error $\mathbb E|\hat p(A=1\mid X) - p(A=1\mid X)|$ in Theorem 2?  In fact, regularization could in fact be reducing the aforementioned error despite huring accuracy.  One way to measure this is, e.g., using reliability diagrams.  The conclusion in D.2 that "this further confirms the robustness of our post-processing modification algorithm" does not make sense.

1. Some clarifications would be helpful:

	- Example 3 does not imply subgroup fairness, i.e., intersecting groups.
	- When introducing the composite criterion, it is also useful to mention that some fairness criteria are incompatible with each other (e.g., DP vs. EO).

1. Related work on the Bayes optimal fair classifier in the attribute-aware setting (via post-processing) are missing, e.g., [1, 2, 3, 4].

[1] Denis et al. Fairness guarantee in multiclass classification. 2023.
[2] Zeng et al. Bayes-Optimal Classifiers under Group Fairness. 2022.
[3] Gaucher et al. Fair learning with Wasserstein barycenters for non-decomposable performance measures. AISTATS 2023.
[4] Xian et al. Fair and Optimal Classification via Post-Processing. ICML 2023.

**Questions:**

See weaknesses.

---

> ### Author Response · Authors · 2023-11-16
> **Response to Reviewer v53s**
>
> Thank you for your review and valuable suggestions on our work. We would like to address your questions/concerns below.
>
> > Q1. It is not mentioned how nonbinary sensitive attributes are handled. Related: How is the sensitive attribute of "race" in the COMPAS experiments, which can be one of three categories (African-American, Caucasian, Other), handled?
>
> Similar to what has been done in existing papers, such as Zafar et al. 2018 and Cho et al. 2020, we preproposess sensitive attribute 'race' in this dataset such that A = 1 represents ‘African American’ and A = 0 corresponds to all other race.
>
> > I am skeptical about the scalability of the proposed method to large datasets and more sensitive attributes. It appears that the time complexity would scale exponentially in the number of sensitive attributes, i.e., $M^K$. I see that $M$, the number of samples on which the decision boundaries are considered, is set to no more than 5000 in the experiments. But practical ML datasets nowadays can contain tens of thousands to millions of examples.
>
> As a post-processing method, we already have much better scalability than in-processing methods in the sense that we don't need to train multiple models to obtain different accuracy-fairness tradeoffs. We only need to train a base model once and run the post-processing algorithm to obtain all different trade-offs. In all our experiments, the time cost for running the post-processing algorithm is much smaller than training the base model, especially for experiments on CelebA where a deep ResNet is trained as the base model.
> Some post-processing methods are indeed quite fast, but they often only provide a single solution targeting at the strict fairness constraint ($\delta=0$). While our post-processing algorithm only needs to be run once to obtain all different trade-offs (corresponding to different desired fairness constraints $\delta's$). Furthermore, we provide additionally a more efficient alternative algorithm (Algorithm 3) in Appendix B, which scales according to $N_{val} \log N_{val}$ when EO is considered
>
> > Q3. Theorem 2 is very important as it provides fairness guarantees for the classifier obtained through the procedure. But the result looks wrong to me, and seems to have a discrepancy with Algorithm 1. The proof is also very hard to read, containing several typos.
>
> Thank you for this suggestion. The result indeed can be extended to general composite criterion (CC) with the same technique, apart from the continuity bound on $Acc^*(\delta) - Acc^*(\delta - \epsilon)$ in the end of the proof, which now requires a slightly different argument. We have updated the proof in the revision to include general CC criterion, and changed the formulation in the main text. We also added some outline at the beginning of the proof, with the hope that it improves readability. Regarding the discrepancy with Algorithm 1, we believe there is a misunderstanding. When we choose $M = N_{val}$, Algorithm 1 and Algorithm 2 go through all possible linear rules on a given validation set, therefore correspond exactly to the procedure described in theoretical formulation. When $M < N_{val}$, we simply reduce the search space, while EO and ACC are still evaluated on the whole validation set. This is done as a heuristic, for sake of reducing the compute time.
>
> We want to stress that Theorem 1 is the main result of our paper, which also gives raise to a practical algorithm. Although Theorem 2 is important from statistical learning point of view, we did not have the goal to have the most tight bound, rather include the result as a sanity check.

---

> ### Author Response · Authors · 2023-11-16
> **Response to Reviewer v53s (part 2)**
>
> > Q4. In the experiments, the authors compared their proposed post-processing algorithm to ones that are attribute-aware, but their algorithm is run in attribute-unaware mode. The authors should have compared to those algorithms by running their algorithm in the same attribute-aware mode. In this sense, the current set of experiments is incomplete.
>
> In Remark 2.1, we show that when run in sensitive attribute-aware mode, our method reduces a group-aware thresholding. However, the method is proposed to be run in sensitive-unaware mode: the fact that our method is run in sensitive attribute-unaware mode and can still achieve competitive or better performance compared with sensitive attribute-aware methods demonstrates the exact advantages of our method that we intend to empirically verify: (i) it doesn't require inference time sensitive attribute, (ii) in some cases, it can still outperform methods requiring inference time sensitive attribute.
>
> > Q5. The conclusions drawn from the ablation study in section D.2 do no make sense to me. How is accuracy related to the error $E[|\hat{p}(A=1|X)-p(A=1|X)|]$ in Theorem 2? In fact, regularization could in fact be reducing the aforementioned error despite huring accuracy. One way to measure this is, e.g., using reliability diagrams. The conclusion in D.2 that "this further confirms the robustness of our post-processing modification algorithm" does not make sense.
>
> Since we don't know the ground-truth $p(A|X)$, there is no way for us to compute $E[|\hat{p}(A=1|X)-p(A=1|X)|]$, and thus this assumption is made purely for theoretical analysis of bounds for performance (which may not be tight). For this ablation, we purely adjust weight decay to obtain 10 different sensitive conditionals $\hat{p}(A|X)$. Within this set of estimated conditionals, some of them are closer to ground-truth and some are not. While we can not tell which model is closer to the ground-truth, the fact that our method yields good performance for all of these 10 different $\hat{p}(A|X)$ shows the robustness of the method. Indeed, higher accuracy does not imply better calibration. However, our theoretical result requires individual/conditional calibration, which is much stronger than marginal calibration and, as mentioned above, is difficult to assess. Reliability diagrams and calibration error based metrics, such as ECE, are known to be problematic when assessing conditional calibration [1, 2]: trivial solutions can obtain perfect marginal calibration (ECE=0), e.g. in a binary classification problem with balanced labels, predicting $\hat{p}(Y=1|X)=0.5$ all the time no matter what $X$ is.
>
> [1] Yuksekgonul et al. Beyond Confidence: Reliable Models Should Also Consider Atypicality. 2023.
>
> [2] Xiong et al. Proximity-Informed Calibration for Deep Neural Networks. 2023
>
> > Q6. Some clarifications would be helpful. Example 3 does not imply subgroup fairness, i.e., intersecting groups. When introducing the composite criterion, it is also useful to mention that some fairness criteria are incompatible with each other (e.g., DP vs. EO).
>
> Indeed, Example 3 does not imply parity for all 4 intersections, rather parity for two pairs. If we want to equalize positive outcome in all 4 groups, such case is equivalent to a multi-label sensitive attribute. In this case, one solution is to compare every pair among the 4 values (which accounts for 6 restrictions). For instance, if $A$ takes values $1, 2, 3, 4$, we can set $k = 6$, $A_1, \dots, A_6 = A$, $(a_1, b_1) = (1, 2)$, $(a_2, b_2) =(1, 3)$, $(a_3, b_3) =(1, 4)$, $(a_4, b_4) =(2, 3)$, $(a_5, b_5) =(2, 4)$, $(a_6, b_6) =(3, 4)$ in Eq(4) on page 3.
>
> We also note that although EO and DP can be at odds, they are not exactly incompatible since constant classifier satisfies both. In fact, constant classifier satisfies any such condition CC = 0. The incompatibility happens when we add Predictive Parity into the mix [3]. The latter is not covered by our composite criterion.
>
> [3] A. Chouldechova (2017) Fair Prediction with Disparate Impact: A Study of Bias in Recidivism Prediction Instruments
>
> > Q7. Related work on the Bayes optimal fair classifier in the attribute-aware setting (via post-processing) are missing.
>
> A: Thank you for pointing out these related works. We will include them in future revision.

---

> > ### Comment · Reviewer_v53s · 2023-11-22
> >
> > I thank the authors for the response, however, most of my concerns remain...
> >
> > Q2.  Indeed, the running time for post-processing is much less than in-processing because we are not training the classifier from scratch, but the proposed algorithm has an exponential dependency on the number of constraints.  Although the number of constraints is always bounded if the authors restrict to binary classification with binary sensitive attribute and consider only the fairness criteria of DP, EO, this weakness generally remains if one were to impose arbitrary number of constraints.
> >
> > Q3.  While I indeed appreciate the main result of Theorem 1, I do not think the importance of Theorem 2 should be downplayed.  This result is quite standard in post-processing literature (e.g., both sample complexity and sensitivity analyses can be found in the papers I references), and to be frank, if this theorem were absent from the paper, I would have given a lower score.  The performance of post-processing, both in fairness guarantee and suboptimality, depends on the accuracy of the predictors $Y\mid X$, $A\mid X$, $(A,Y)\mid X$ in terms of the $L^1$ error.  In practice, it is unlikely that one will get an exactly accurate predictor, so it is important that we provide the sensitivity analysis to inform the practitioner how (un)fair (or suboptimal) will the classifier obtained from the post-processing algorithm be given the predictors that they have.
> >
> > - Re. the dependency on $M$.  Would it mean that, when $M\leq N_\mathrm{val}$, the sample complexity would depend on $M$ and not $N_\mathrm{val}$?  If so, then it circles back to my concern regarding the efficiency, since in your experiments, $M\leq5000$ is quite small.  Can you scale to a larger $M$?  Also, how long would it take to run?
> >
> >     The goal is not to have the tightest bound, but simply a correct bound.
> >
> > - The result still looks wrong.  DP (and EO) should depend on $\epsilon$—if the predictor of $A\mid X$ is wrong, then how can the algorithm produce a fair classifier?  This indicates potential errors in the proof...
> >
> > Q4.  I would imagine running the proposed algorithm in both attribute-aware and unaware modes to compare to existing in-processing (where some may be unaware) and post-processing algorithms (where most are aware), for fair comparison, since they target both settings.
> >
> > Q5.  I do not understand why the "robustness of our post-processing modification algorithm" can be confirmed by a set of experiments showing that the accuracy of $A\mid X$ does not effect the post-processing, since the algorithm does not use $\argmax A\mid X$ but the $[0,1]$ likelihood.  What does robustness mean here?  It should mean being robust to the $L^1$ error of $A\mid X$.  Then the authors should have measured that instead of accuracy.
> >
> > Q6.  In this case, please clarify the wording in the paper.  The current phrasing of "fairness with respect two sensitive attributes A, B simultaneously (for instance, gender and race)" may be mistaken for subgroup fairness.
> >
> > I appreciate the authors' detailed response, however, given that most of my concerns and especially that the issue with theorem 2 remains, I am inclining to lower my score.

---

> ### Author Response · Authors · 2023-11-22
>
> We thank the reviewer for the response and would like to address the issues one more time.
>
> Q3 & Q5.
> > the dependency on $M$
>
> Indeed, the result only holds for $M = N_{val}$. We will emphasise this clarification in the final revision in the main text. We note however, that in Algorithm 2 both $M$ and $N_{val}$ play a role. It's not like we are replacing the sample size with $M$. The value $M$ only affects the search of $\hat{\gamma}$ in Eq. 17 in the same way SGD approximates ERM (for sake of speed). In Algorithm 2, we still use the whole validation sample ($N_{val}$) for calculating the empirical statistics $\widehat{Acc}$ and $\widehat{EO}$. The sampling error comes from the uniform inequalites (end of p. 25) which depend on the sample size ($N_{val}$, not $M$).
>
> > The result still looks wrong. DP (and EO) should depend on —$\epsilon$ if the predictor of  is wrong, then how can the algorithm produce a fair classifier?
>
> Firstly, if the error $\varepsilon$ is too bad, then Eq. 18 will not hold, so we do not have guarantees when $A|X$ is too bad. The conditions still require that $\varepsilon$ is small enough and $N$ is large enough. However, this does not mean the error of CC have to depend on $\varepsilon$. Let us bring up some relevant details from the proof.
>
> From our definition Eq. 17, we have to have that $\widehat{CC}(\check{Y}_{\hat{\gamma}}) \leq \delta $.
> The uniform inequalities (beginning of p. 26) are pretty standard application of VC inequality. In particular, they say that
>
> $\widehat{CC}(\check{Y}_{\gamma} )$
>
> and
>
> $CC(\check{Y}_{\gamma} )$
>
> are close for ALL $\gamma$, including $\hat{\gamma}$ if it exists. The difference between them does not depend on $\varepsilon$. In order to show that $\hat{\gamma}$ exists (see 3 lines below Eq. 22) we have to assume Eq. 18, and now we do need to have a small $\varepsilon$.
>
> > showing that the accuracy of $A|X$  does not effect the post-processing
>
> The experiment in Figure 3 shows approximately that. There, instead we directly ``poison'' the model by adding random noise independently for each $X$. This means that with increasing alpha we increase the $E | \hat{p}(A|X) - p(A|X)|$ by $const \times \alpha$. We also want to draw your attention to Figure 3 (b), where the DP does not steer away too much when we increase alpha. This experimentally  confirms that the error of DP in Theorem 2 does not have to depend on $\epsilon$!
>
> > Q6. In this case, please clarify the wording in the paper.
>
> Thank you for this suggestion, we will add this clarification in the final revision.
>
> > Q4. I would imagine running the proposed algorithm in both attribute-aware and unaware modes to compare to existing in-processing (where some may be unaware) and post-processing algorithms (where most are aware), for fair comparison, since they target both settings.
>
> In the sensitive aware mode, i.e. when we assume that $p(A|X)  \in$ { 0, 1}, our method turns into sensitive-dependent thresholding (as we show in Remark 2.1). These kind of methods has been extensively studied in the literature, so we believe it is not necessary to do it again. We are happy to include the list of relevant literature that you kindly provided in the original review in a "relevant literature" section when we complete the final revision.

---

> ### Comment · Reviewer_v53s · 2023-11-22
>
> DP in theorem 2 has to generally depend on $\epsilon$.
>
> - Suppose $X=\\{0,1\\}$ uniform, and $A = 1[X=0]$.  Let $Y=1[X=0]$ be known.
> - The unconstrained optimal classifier is $\widehat Y=1[X=0]$, which is not fair, because $\widehat Y=1[A=1]$.
> - If $\widehat A\mid X$ is wrong, e.g., constant $\widehat A=1/2$, then the "optimal fair" classifier based on $\widehat A$ is $\widehat Y=1[X=0]$, which is not fair.

---

> ### Author Response · Authors · 2023-11-22
>
> There are a number of issues with your example which we highlight below.
>
> Firstly, if the estimate is a constant, that will correspond to a high estimation error, i.e., a high epsilon. In this case, the condition we assumed in Eqn. 18 will not satisfy and therefore our results do not apply or do not conflict with the example.
>
> In short, the DP does depend on the epsilon, but in an implicit way where the underlying assumption assumes the estimation error has to be bounded and be moderate and it will interact with the other parameters through the constraint.
>
> We have been explicit about this in our paper: at the beginning of page 23, right below Theorem 2:
> "Before we move on to the proof, let us briefly comment on each condition. The first condition
> requires that we have a good estimation of probabilities in the form of moment bound."
> as well as in our previous rebuttal.
>
> Secondly, we also want to highlight that we have stated in our theorem and the theorem is true for continuous variables $X$, and the  ground-truth scores $f(X)/\eta(X)$ has continuous distribution. This assumption is explicitly stated in the main statements of both Theorem 1 & 2.
>
> Thirdly, we do not see a validation sample in your example. The fairness of the modified estimator $\check{Y}$ by definition assessed by an observed validation sample $(X, Y, A)$ which comes from the true distribution (not the erroneous $\hat{A}|X$).
>
> We hope this clarifies but would be happy to discuss further.

---

> > ### Comment · Reviewer_v53s · 2023-11-23
> >
> > I thank the authors for the response.
> >
> > - In this case, please clarify what is meant by "This experimentally confirms that the error of DP in Theorem 2 does not have to depend on $\epsilon$!" in the previous response, since in the newest response it is said that "DP does depend on the epsilon".
> > - It is very misleading to simply hide $C$ in the informal version of the theorem on page 7 using $\lesssim$; please make explicit the requirement/interaction on both $N$ and $\epsilon$.

---

> ### Author Response · Authors · 2023-11-23
>
> Thanks for the response.
>
> * "DP" does depend on $\epsilon$ " means if the condition (Eq. 18) is not satisfied (e.g. when $\epsilon$ is too big. In this case, our algorithm shouldn't work), then our bound doesn't hold. When Eq. 18 holds, then the problem in Eq. 17 for $\hat{\gamma}$ is feasible, in this case, the effect of $\epsilon$ will be dominated by other factors and thus doesn't show up in the bound. So "This experimentally confirms that the error of DP in Theorem 2 does not have to depend on $\epsilon$" means Fig. 3 (b) empirically verifies that when Eq. 18 is satisfied (implying moderate $\epsilon$), the effect of $\epsilon$ doesn't play an important role in error of DP, which is compatible with why $\epsilon$ doesn't show up in the bound for DP under the underlying assumption.
>
> * Thank you for the suggestion, we will include more clarifications of $C$ in the Informal version of Theorem 2 in the main text in next version.

---

### Official Review · Reviewer_cdR8 · 2023-10-25

**Soundness:** 3 good
**Presentation:** 3 good
**Contribution:** 3 good
**Rating:** 8
**Confidence:** 3

**Summary:**

The paper characterizes the optimal classifier under fairness constraints as a simple postprocessing modification rule over the Bayes optimal classifier. Comparison with standard baselines demonstrates competitive results on three datasets.

**Strengths:**

- Novel characterization of the optimal fairness-constrained classifier as a "simple" modification rule over the Bayes optimal classifier.
  - Group-specific thresholding (Hardt et al., 2016) is a specific case of this rule where the sensitive attribute data is known at inference time; proving that simple thresholding is optimal for DP and EO when this information is known (with Bayes optimal scores).
  - Specific examples given for DP, EO, and equalized odds.
- The proposed method does not need explicit access to the sensitive attributes at inference time, but can also be given this information if available.
- Experiments conducted with relevant baselines on three well-known datasets, supporting the main paper claims.
- Additional sensitivity analysis and ablation studies on the robustness of the method to miss-estimated $p(A|X)$ or $p(Y|X)$.

**Weaknesses:**

- No code or results files are provided for the experiments; neither an implementation for the proposed method. This is largest point against the current version of the paper, as properly reviewing the work required checking some experimental details.

- Given that postprocessing baselines achieve Pareto dominant results in Fig. 2 (expectedly, as they have access to the sensitive attribute at inference time), it would be interesting to add partially relaxed results for these baselines for a more direct comparison (as done for the Zafar method).

Some comments regarding the CelebA results on Table 1:
- The proposed method is fitted with relaxed fairness constraint fulfillment ($\delta > 0$), while baselines are not ($\delta=0$). This does not seem to be a completely fair comparison.
- I'd find the small metric differences more meaningful if the "bolded results" rule were based on pair-wise statistical significance tests.
  - e.g., the bolded results of Table 5 are perhaps not significant.

Other notes:
- The compatibility with multiple over-lapping sensitive sub-groups (Example 3) is definitely a major advantage, but no experiments are shown for this evaluation setting.
- It'd be interesting to test against a simple baseline of using Hardt et al. group-specific thresholding using the same estimated $p(A|X)$ instead of the true sensitive attributes at inference time.

**Questions:**

- Is the base model used by MBS the same as those used by the baselines? Are the Zafar et al. (2017) results of Fig. 2 based on a constrained MLP?
- How was $p(Y,A|X)$ estimated when using MBS on CelebA?
- Do you see any reason why Hardt et al. (2016) would outperform on the Fig. 2 results, and achieve such lacklustre results on Table 1? Given that we see some variance/unreliability on fairness for MBS with $\delta=1$, can the even stricter constraint target by Hardt et al. (2016) (which uses $\delta=0$, right?) be related to its underperformance?
- Could you please clarify the main differences to Zeng et al. (2022), as it seems to tackle exactly the same problem.
> [Zeng, Xianli, Edgar Dobriban, and Guang Cheng. "Bayes-optimal classifiers under group fairness." arXiv preprint arXiv:2202.09724 (2022).]

Minor:
- Ticks for horizontal axes in Figures 3, 4 and 5 are miss-labeled.
  - Also, clarify that corrupted $p(Y|X)$ is the left figure, and $p(A|X)$ the right figure in the legend or plot titles.

---

> ### Author Response · Authors · 2023-11-16
> **Response to Reviewer cdR8**
>
> Thank you for your encouraging review and your valuable suggestions on our work. We will address your questions below.
>
> > Q1. No code or results files are provided for the experiments; neither an implementation for the proposed method. This is largest point against the current version of the paper, as properly reviewing the work required checking some experimental details.
>
> Details of hyperparameters and model architectures can be found in the beginning of Section 4. We will release the code in the case of acceptance, at the moment we are still tidying up. We will include the uncleaned version of the code in an anonymous github repo and provide the link by rebuttal deadline.
>
> > Q2. Given that postprocessing baselines achieve Pareto dominant results in Fig. 2 (expectedly, as they have access to the sensitive attribute at inference time), it would be interesting to add partially relaxed results for these baselines for a more direct comparison (as done for the Zafar method).
>
> This is an interesting point. However, the two post-processing baselines considered in current version of the paer doesn't allow this relaxation: Wang et al. 2019 (the post-processing baseline for DP) is designed to enforce independence between the prediction and sensitive information by minimizing Wasserstein-1 distance, and Hardt et al. 2016 (the post-processing baseline for EO) is designed to satisfy the strict EO constraint (i.e., targeting at achieving 0 EO). Thus, there is no way to relax the constraints in these methods. We will leave the comparison between MBS and other post-processing methods that allow relaxation in future work (any suggestions of such baselines are very welcome).
>
> > Q3. The proposed method is fitted with relaxed fairness constraint fulfillment ($\delta$>0), while baselines are not ($\delta$=0). I'd find the small metric differences more meaningful if the "bolded results" rule were based on pair-wise statistical significance tests.
>
> As mentioned in our response to Q2, many post-processing baselines are proposed to try to satisfy the strict constraint (i.e. disparity=0), thus they are less flexible than MBS which treats the desired level of constraint $\delta$ as a hyperparameter.
> We include the standard error of the experimental results in Appendix D, which comes from training for 3 seeds. Significance testing is usually concerned with sampling error, but for the Celeb-A datasets the splits are fixed in the original dataset. Similar error bars are reported in previous papers that use Celeb-A benchmark.
>
> > Q4. Is the base model used by MBS the same as those used by the baselines? Are the Zafar et al. (2017) results of Fig. 2 based on a constrained MLP?
>
> Yes, they are the same. The results of Zafar et al. (2017) in Fig. 2 are based on constrained MLP with architecture exactly the same as the base model for post-processing methods.
>
> > Q5. How was p(Y,A|X) estimated when using MBS on CelebA?
>
> We trained ResNet-18 with 4 classes corresponding to (Y=0 or 1, A=0 or 1) using cross-entropy loss. The softmax of the logits is then interpreted as $\hat{p}(Y,A|X)$. These experimental details can be found at the beginning of Section 4.
>
> > Q6. Do you see any reason why Hardt et al. (2016) would outperform on the Fig. 2 results, and achieve such lacklustre results on Table 1? Given that we see some variance/unreliability on fairness for MBS with $\delta$=1, can the even stricter constraint target by Hardt et al. (2016) (which uses $\delta$=0, right?) be related to its underperformance?
>
> A: From our theoretical result, our method will be close to optimal if the base model is close to Bayes-optimal classifier and the estimated conditionals are close to ground-truth, thus it does have potential to outperform Hardt et al. (2016). Besides, Hardt et al. (2016)'s modification rule is oblivious (i.e., it occurs at the group level while ignoring the details of individual feature X), while our method bypass this limitation since $\check{Y}(X)$ takes advantage of individual feature X.  Furthermore, indeed Hardt et al. (2016) targets at strict fairness constraint $\delta=0$, which may hurt accuracy in order to satisfy such strong constraints.
>
> > Q7. Could you please clarify the main differences to Zeng et al. (2022), as it seems to tackle exactly the same problem.
>
> A: Thank you for pointing out this reference, we were not aware of it. As far as we can tell, they only tackle DP, EOp while we consider additionally EO and composite criterion. Furthermore, they study optimality in the class of sensitive attribute aware classifiers, while our classifiers only depend on $X$. We will include this related work in future revision.
>
> > Q8. Ticks for horizontal axes in Figures 3, 4 and 5 are miss-labeled. Also, clarify that corrupted p(Y|x) is the left figure, and P(A|X) the right figure in the legend or plot titles.
>
> A: Thank you for pointing out the typos. We have fixed them in the revision.

---

> > ### Author Response · Authors · 2023-11-20
> > **Response to Reviewer cdR8 (part 2)**
> >
> > We have put the uncleaned version of the code in an anonymous github repo: https://github.com/mbsmbsmbs/MBS/tree/main

---

> > ### Comment · Reviewer_cdR8 · 2023-11-22
> >
> > Thank you for the clarifications.
> >
> > > From our theoretical result, our method will be close to optimal if the base model is close to Bayes-optimal classifier (...) thus it does have potential to outperform Hardt et al. (2016).
> >
> > Theorem 4.5 of Hardt et al. (2016) had already proved this fact for group-dependent thresholding (the postprocessed classifier will be near-optimal if the base model is near-Bayes-optimal, under strict fairness). So I don't think this explains why your method would outperform thresholding. I maintain that the reason why it outperforms seems to be that it is fit for a relaxed constraint fulfillment, which is not entirely a fair comparison. The paper would greatly benefit from having rows showing results for $\delta=0$ for an apples-to-apples comparison.
> >
> > > Hardt et al. (2016)'s modification rule is oblivious (...) while our method bypass this limitation since
> >  takes advantage of individual feature X.
> >
> > When given the ground-truth sensitive attribute information at inference time, is your method not oblivious as well? While the Hardt et al. method assumes access to sensitive attribute, your method is not oblivious in the general case simply because it must estimate the sensitive attribute from $X$. As far as I can tell this method tells us how to optimally join a group-dependent threshold rule together with a model that predicts a sample's sensitive attribute. The most interesting point in my opinion is that a simple thresholding rule seems to not be sufficient when using _relaxed_ equalized odds (it's known to be sufficient for strict equalized odds).

---

> ### Author Response · Authors · 2023-11-22
> **Thank you for the response**
>
> We thank the reviewer for the response and suggestions. We will run the experiments with stricter constraints and add the results into our next version for a complete comparison.

---

### Official Review · Reviewer_ErzV · 2023-11-02

**Soundness:** 3 good
**Presentation:** 3 good
**Contribution:** 3 good
**Rating:** 8
**Confidence:** 4

**Summary:**

This paper considers the problem of fair learning through post-processing: given an arbitrary predictor, we would like to post-process its predictions such that the new predictions satisfy a notion of group fairness, say demographic parity, while maintaining good accuracy. Standard methods for fairness through post-processing come up with a predictor that takes the sensitive attribute as input, and therefore, requires access to sensitive attributes at test time. This is not desirable because, in practice, laws and regulation might prohibit access to sensitive information. This paper introduces a new post-processing method that does not require this access; instead, it works with a conditional distribution of sensitive attributes (conditioned on all other features). More formally, let $(X,A,Y)$ represent features, sensitive attributes, and labels. Given a base classifier $\hat{Y} (X)$, a conditional distribution $\hat{P} (A,Y|X)$, the paper introduces an efficient algorithm that gives us a new classifier $\hat{Y}’ (X)$ that satisfies a desired notion of fairness, while approximately preserving the accuracy of $\hat{Y}$. They experiment with their proposed algorithm on Adult, COMPAS and CelebA data sets and find that in most cases their proposed algorithm outperforms (some) existing fair learning algorithms.

**Strengths:**

-A key challenge in fair learning is access to sensitive attributes. This paper acknowledges the fact that sensitive attributes may not be accessible in practice, and therefore, proposes a post-processing algorithm that does not require such access. To the best of my knowledge, the proposed method is original and has significant impact.

-The authors accompany their theoretical guarantees with an extensive experimental analysis to show the efficacy of their algorithm.

-The paper is well-written and is easy to read.

**Weaknesses:**

-While the proposed method does not require access to sensitive attributes at test time, it still requires the conditional distribution of sensitive attributes $P(A|X)$, or a good estimate of it. It is not clear if this complies with laws and regulations: a company can still use their model of $P(A|X)$ to get good estimates of individual’s sensitive attribute. I’d like to see a discussion of this in the paper as well.

-Overall, the assumption that we have access to $P(Y, A|X)$, or a good estimate of it, could be strong in practice. For example, if I know a good estimate for $P(Y|X)$, I might as well use that as my predictor. Also, how are these conditional distributions learned? In practice, we observe every $x$ only once, so these probabilities are 0 or 1 on observed data, unless we work with parametric models like logistic regression. But which parametric model should we use here when the underlying unknown data distribution could be arbitrary? Also, how are these models chosen in your experiments?

-The paper claims that the performance of their method is better than “in-processing methods”. Is it better than all in-processing methods or just a few? This sounds like a very strong claim because, generally speaking, in-processing methods do achieve better performance than post-processing methods. Additionally, the most popular in-processing method for fair learning is given by Agarwal et al. 2018 (titled: "A reductions approach to fair classification"). Unfortunately, their algorithm is not included in the benchmarks for experiments. I would like to see a comparison of the two methods.

**Questions:**

-Do the theoretical results rely on the fact that $\hat{Y}$ is the Bayes optimal classifier. $\hat{Y}$ is introduced as the Bayes optimal on page 3 but later on is used as any predictor. It would’ve been better if $\hat{Y}$ was initially introduced as any predictor that we’d like to post-process its predictions.

-Can the validation data set be used to learn the conditional distributions? In practice we only have a pre-trained classifier and do not necessarily have pre-trained conditional distributions. If your method allows using the same validation set to learn these distributions, then all you’d need is the pre-trained classifier, increasing the flexibility of the proposed method.

-The title of the paper seems misleading. What does “optimal” mean here? Post-processing algorithms are known to be sub-optimal in general because their guarantees are benchmarked against the base classifier (e.g., see your theorem on page 7). Theoretically, in-processing methods achieve the optimal tradeoff between accuracy and fairness because they directly solve the constrained optimization problem instead of looking at the specific class of models that are derived by post-processing another model. This does need a clarification in the paper.

-Why does Hardt et al. (2016) have lower performance than your proposed method in the experiments? Hardt. et al. (2016) solves the same post-processing problem with the extra flexibility that the sensitive attribute can be used as an input to the model. Shouldn’t that just lead to better accuracy/fairness tradeoff?

-------
I will increase my score if questions/weaknesses discussed above are addressed properly.

---

> ### Author Response · Authors · 2023-11-16
> **Response to Reviewer ErzV**
>
> Thank you for your encouraging review and valuable feedback. Below we address each question one by one
>
> > Q1. Still requires the conditional distribution of sensitive attributes p(A|X), or a good estimate of it. It is not clear if this complies with laws and regulations.
>
> Thank you for raising this question. Indeed, collecting senstive attributes can be prohibited by law, but we didn't manage to find proper references that confirm that using proxies is also regulated. There is indeed raising concern expressed in the literature [1] and in the media [2]. However, since we do not specialize in legal questions, we are not confident mentioning regulations and law in regard of using proxies. We would highly appreciate your help if you could refer us to the correct source. We will mention [1] as a limitation in the final version.
>
> [1] Eduard Fosch-Villaronga. Gendering algorithms in social media
>
> [2] https://www.theguardian.com/technology/2018/may/16/facebook-lets-advertisers-target-users-based-on-sensitive-interests
>
> > Q2. The assumption that we have access to p(Y,A|X) or a good estimate of it, could be strong in practice. How are these conditional distributions learned?
>
> We can fit a model to obtain the estimated conditional distributions $\hat{p}(Y,A|X)$ as long as we have access to a set of observed triplets $(X,A,Y)$, which is assumed for most methods. The decision of what target model $\hat{p}(Y|X)$ to use is up to practitioners: one can either use a pretrained one or obtain it from $\hat{p}(Y,A|X)$ depending on which strategy gives the best target accuracy - but even the best prediction model here might suffer from fairness violations and our post hoc approach can still plug in to improving its fairness property. Indeed, these conditionals are estimated using parametric models. Specifically, in our experiments, we use neural networks with cross entropy loss, which have shown profound success in prediction tasks and fitting conditionals. For each dataset, we use the same neural network architecture for all the methods compared. The base model for post-processing methods and the best model for in-processing methods are selected according to performance over validation set. We also note that, from our ablation study (Appendix C.1, C.2), the auxiliary model for sensitive attribute does not have to be a very good fit.
>
> > Q3. The paper claims that the performance of their method is better than “in-processing methods”... Additionally, the most popular in-processing method for fair learning is given by Agarwal et al. 2018 (titled: "A reductions approach to fair classification")
>
> In the abstract we state that we achieve competitive or better performance, in particular, the performance on Adult and COMPAS is comparable, while on Celeb-A we outperform the state-of-the-art method Park et al (2022), which is concerned specifically with this vision probelem. In the experiments we always use the same base model.
>
> As per request, we compare to the method provided by Agarwal et al. Due to the time constraint of rebuttal, we only run it for Adult Census dataset for now, but we will run the full set of experiments and include the results in the future. Since MLPClassifier in sci-kit learn does not have option sample_weight required by the package (fairlearn) supporting reduction method, we implement our own pytorch version, with same hyperparameters as for other baselines. It appears reduction approach does not perform very well with MLP, in particular, the results below were selected using the official selection script https://fairlearn.org/v0.5.0/auto_examples/plot_grid_search_census.html:
> For DP constraints:
> | Acc | DP |
> | --- | --- |
> | 0.7570 | 0.0067 |
> | 0.8056 | 0.0082 |
> | 0.8118 | 0.0279 |
> | 0.8184 | 0.0658 |
> | 0.8189 | 0.0883 |
> | 0.8221 | 0.1191 |
>
> For EO constraints:
> | Acc | EO |
> | --- | --- |
> | 0.8182 | 0.1002 |
> | 0.8218 | 0.1479 |
> | 0.8228 | 0.0446 |
>
> Please also see Figure 6 in Appendix F in the revised paper where we plot the performance of Agarwal et al. along with the other methods.
>
> > Q4. Do the theoretical results rely on the fact that  $\hat{Y}$ is the Bayes optimal classifier. $\hat{Y}$ is introduced as the Bayes optimal on page 3 but later on is used as any predictor. It would’ve been better if $\hat{Y}$ was initially introduced as any predictor that we’d like to post-process its predictions.
>
> Yes, the theoretical result is based on the assumption that $\hat{Y}$ is the Bayes-optimal classifier, see e.g. introduction p 2 paragraph 3, or beginning of section 2. However, MBS, as a practical method, itself does not restrict us with the choice of base classifier, so we think of $\hat{Y}$ as any score-based estimator that was fit without fairness constraints.

---

> ### Author Response · Authors · 2023-11-16
> **Response to Reviewer ErzV (part 2)**
>
> > Q5. Can the validation dataset be used to learn the conditional distributions? In practice we only have a pre-trained classifier and do not necessarily have pre-trained conditional distributions. If your method allows using the same validation set to learn these distributions, then all you’d need is the pre-trained classifier, increasing the flexibility of the proposed method.
>
> In principle, it can. However, we note that in practice the classifiers are often score-based, which can be interpreted as conditional probability. As we mention in our response to Q2, we can learn the conditional distribution as long as we have access to a set of observed triplets $(X,A,Y)$. We also note that in some cases, we can use a zero-shot classifier as we do in section C.3. In that regard, MBS is a very flexible approach.
>
> > Q6. The title of the paper seems misleading. What does “optimal” mean here?
>
> The optimality is concerned with the case where we know the ground truth distribution, and impose the group-fairness constraints (as mentioned in paragraph 3 in page 2.). It turns out that such (constrained) optimal classifier can be seen as modification of the unconstrained Bayes-optimal classifier $\hat{Y} = 1[p(Y = 1|X) > 0.5]$. This question is fundamental and before the solution was only available for simple constraints (DP & EOp), see Menon & Williamson (2018).
>
> > Q7. Why does Hardt et al. (2016) have lower performance than your proposed method in the experiments?
>
> Hardt et al. (2016)'s modification rule is oblivious (i.e., it occurs at the group level while ignoring the details of individual feature X), while our method bypass this limitation since $\check{Y}(X)$ takes advantage of individual feature X.  Furthermore, Hardt et al. (2016) target at strict fairness constraint $\delta=0$, which may hurt accuracy in order to satisfy such strong constraints.

---

> > ### Comment · Reviewer_ErzV · 2023-11-22
> > **Response**
> >
> > Thank you for your thorough response -- I'll raise my score.
> > My only comment is regarding optimality: because we never know the ground truth distributions, in-processing methods when we fix a class of models, at least theoretically, give the optimal tradeoffs between fairness and accuracy for that class. When the sensitive attributes A are not available at test time, such methods can easily be restricted to output a model that takes only features X as input.

---

> ### Author Response · Authors · 2023-11-22
> **Thank you for the response**
>
> We thank the reviewer for the follow up. We will add a discussion in future version about the in-processing method's optimality (within the specified model class) and clarify that indeed these methods can be restricted to models that take only X as input during inference.

---

### Official Review · Reviewer_in91 · 2023-11-08

**Soundness:** 3 good
**Presentation:** 2 fair
**Contribution:** 3 good
**Rating:** 8
**Confidence:** 4

**Summary:**

This paper deals with the problem of fair classification where the goal is to find the classifier with maximum possible accuracy under constraints on the disparity in the performance of the classifier across groups with different values for some protected attributes. The paper proposes a post-hoc approach to achieve this.

For binary classifiers: First, an unconstrained Bayes optimal classifier (Y'(X)), which maximizes accuracy, is learned. Then post-hoc, a modification rule is used to obtain a fairness constrained Bayes optimal classifier (Y''(X)) by modifying the output of Y'(X). This modification is done by mapping each instance to a probability with which the fairness constrained classifier disagrees with the unconstrained classifier.

The paper proposes a definition of such a modification rule which is defined by an instance-level bias score which the authors propose, together with a measure of the uncertainty of the unconstrained classifier on a given instance.

The authors propose definitions of the bias score for each of three popular fairness constraints, and show how the resulting modification rules leads to classifiers that satisfy the fairness constraints. The authors point out that unlike previous works, their approach enables us to find classifiers satisfying Equalized Odds fairness constraints.

**Strengths:**

- The paper proposes a novel way to modify the output of the unconstrained Bayes optimal classifier post-hoc in order to satisfy fairness constraints. While this approach has been previously studied, I believe the instance-level bias scores are novel.
- The main significant technical contribution is the ability to satisfy Equalized Odds fairness constraints.
- Besides these, the characterization of the optimal modification rule in Theorem 1, which has the form of a linear combination of bias scores, one for each protected attribute is also very interesting. In particular, this enables the approach in Section 3 where together with an auxiliary model that estimates the values for the protected attributes, the bias score for examples in the test set can be computed without access to the values of the protected attributes.
- Together, I think the conceptual and technical contributions are both interesting and significant, and the topic is clearly relevant to ICLR and the research community working on fairness in ML.

**Weaknesses:**

- No major weakness apart a few issues with the writing and minor typos that can be fixed with a revision.

**Questions:**

None

---

> ### Author Response · Authors · 2023-11-16
> **Response to Reviewer in91**
>
> Thank you for your encouraging review. We will read through and fix the typos in final revision.

---

### Author Response · Authors · 2023-11-16
**Revision and rebuttal**

We thank the reviewers for valuable feedback and suggestions. Taking the reviews into account, we submitted a revised version, with changes in the main text highlighted in blue. Here is a list of changes:

1. Fixed typos in the axes in Figure 3,4,5 (raised by Reviewer cdR8).

2. Updated informal Theorem 2 statement (sensitivity analysis) in Section 3 to include the EO case. Updated formulation of Theorem 2 (sensitivity analysis) in the appendix and its proof, to account for general composite criterion case, including EO. (raised by Reviewer v53s).

4. We moved "Sensitivity analysis" section to the end of the appendix, to avoid mixing with algorithms. Because of this the ordering of the sections changed (now ablation study in section C, tables in section D, sensitivity analysis in section E).

5. We included the comparison with reduction method (Agarwal et al. (2018)) on Adult Census dataset in Figure 6, Appendix F (raised by Reviewer ErzV).

We respond to each individual reviewer's questions below.

---

### Meta-Review · Area_Chair_sJxj · 2023-12-11

**Metareview:**

The paper studies how to modify the output of the unconstrained Bayes optimal classifier to satisfy fairness constraints. The reviewers all agree that the results are nice and novel, with solid theoretical and empirical results. The paper is also well-written and should be a nice contribution to the conference. The authors should revise the paper according to the discussion. For example, they should include a discussion on in-processing methods.

**Justification For Why Not Higher Score:**

This is a nice contribution to the fair ML literature. There is still a lot of similarity to established methods. I am not sure this should be viewed as a top submission.

**Justification For Why Not Lower Score:**

This paper seems stronger than a typical ICLR accepted paper.

---

### Decision · Program_Chairs · 2024-01-16

Accept (spotlight)